# Seeing Better and Going Deeper in Cancer Nanotheranostics

**DOI:** 10.3390/ijms20143490

**Published:** 2019-07-16

**Authors:** Maharajan Sivasubramanian, Yao Chen Chuang, Nai-Tzu Chen, Leu-Wei Lo

**Affiliations:** 1Institute of Biomedical Engineering and Nanomedicine, National Health Research Institutes, Zhunan 350, Taiwan; 2Department of Cosmeceutics, China Medical University, Taichung 40402, Taiwan; 3Department of Biological Science and Technology, China Medical University, Taichung 40402, Taiwan

**Keywords:** nanoparticles, clinical diagnosis, cancer imaging and therapy, image-guided therapy

## Abstract

Biomedical imaging modalities in clinical practice have revolutionized oncology for several decades. State-of-the-art biomedical techniques allow visualizing both normal physiological and pathological architectures of the human body. The use of nanoparticles (NP) as contrast agents enabled visualization of refined contrast images with superior resolution, which assists clinicians in more accurate diagnoses and in planning appropriate therapy. These desirable features are due to the ability of NPs to carry high payloads (contrast agents or drugs), increased in vivo half-life, and disease-specific accumulation. We review the various NP-based interventions for treatments of deep-seated tumors, involving “seeing better” to precisely visualize early diagnosis and “going deeper” to activate selective therapeutics in situ.

## 1. Introduction

Light is an electromagnetic radiation that has intrinsic energy depending on its wavelength or frequency. Hence, the use of light as a diagnostic and therapeutic tool in clinics is unparalleled, especially in cancer [1,2,3,4,5,6]. Cancer is one of the most debilitating diseases, causing over 7 million deaths worldwide each year, and threatens to increase in the future [7]. The survival of cancer patients mostly depends on accurate early diagnosis followed by suitable therapeutic intervention [8,9]. Most clinical diagnostic imaging modalities, such as optical imaging (OPI), magnetic resonance imaging (MRI), photo acoustic imaging (PAI), positron emission tomography (PET), single-photon emission computed tomography (SPECT), and computed tomography (CT), are based on the interactions of light with tissue or body fluids [10,11,12,13,14,15,16,17]. Salient features of various clinically available imaging modalities are listed in Table 1. For instance, in fluorescence imaging, light from an external source is absorbed by the imaging agents and instantly reemitted as longer wavelength light. The vital information gained by these interactions assists clinicians in planning suitable treatments and mitigates disease severity. Light-based therapeutic interventions for cancer include photodynamic (PDT) and photo thermal therapy (PTT). The former uses light of a suitable wavelength from an external source to activate a light-sensitive agent, a photosensitizer (PS), which transfers its excited state energy to nearby oxygen molecules to generate cytotoxic free radicals. In PTT, photon energy is converted into heat to kill cancer cells [18,19,20,21].

Vast advances in chemical and biological sciences resulted in the development of various therapeutic and imaging contrast agents. For instance, PET imaging can locate highly proliferative tumors using fluorodeoxyglucose ([^18^F]FDG) [22]. When administered, they can preferentially enter high metabolic rate cells and are phosphorylated, thereby accumulating at the cancer site. Similarly, PAI can image B7-H3 overexpressing breast cancer using affibody conjugated clinically approved near infrared (NIR) dye indocyanine green [23]. Optical imaging fluorescent dyes that can emit strong fluorescence signals in the presence of pathogenic stimuli were developed such as pH probes, oxygen sensitive probes, and bioreductive and activity-based probes [24].

The clinical utility of most of these payloads (imaging and therapeutic agents) suffered setbacks due to their inability to negotiate several critical factors. First, rapid elimination from body circulation by the immune system exists and thus constitutes poor bioavailability. Second, there is a lack of tumor-targeting, which results in sporadic bio-distribution profiles, causing undue toxicity to vital organs. Third, there is an accumulation of an insufficient number of agents in tumors to observe any significant diagnostic or therapeutic effect [25,26,27]. However, nanoparticles (NP) hold great promise to address these limitations with their unique capabilities. For example, the large surface area of NPs allows them to carry and deliver high payloads, which will amplify the desired effect. Selective delivery of payloads in the tumor is also possible by either the enhanced permeability and retention effect or active binding to cancer cells by a ligand-receptor mechanism. Moreover, in vivo short half-lives of payloads can be significantly extended by masking the active NP surface. Owing to these features, enthusiasm for NP as a delivery system has facilitated a new frontier in the biomedical field [28,29,30]. This review article highlights encouraging research directions regarding how NPs can be utilized as a diagnostic imaging tool to “seeing better” the onset of malignant tumors and various strategic “going deeper” approaches to targeting and treating tumors. Finally, we discuss recent developments in the applications of NP as image-guided therapy (IGT) agents for cancer treatment and present general conclusions.

## 2. Seeing Better

### 2.1. Magnetic Resonance Imaging (MRI)

MRI is a noninvasive, nonionizing modality providing refined anatomical, physiological, and molecular information of living subjects with high spatial resolution (~1 mm). The principle of MRI is defined as protons in the presence of a magnetic field being excited by absorbing radiofrequency (RF) energy and relaxing back to their equilibrium state by emitting the absorbed energy, producing an MRI signal [11,31,32]. However, the low sensitivity of MRI prompted the use of NP-based contrast agents, which improves the relaxation rates of water protons even with micromolar or nanomolar concentrations. There are two types of MR imaging mechanisms: *T*1-weighed and *T*2-weighed. *T*1 contrast agents produce bright MR images by longitudinal relaxation of water protons, and *T*2 contrast agents produce dark MR images by transverse relaxation of water protons [33,34,35,36,37].

#### 2.1.1. T1 Contrast Agents

Gadolinium ions (Gd^3+^) are the traditionally used excellent *T*1 contrast agent due to their ability to generate bright signals by shortening the *T*1 time (Figure 1), and hence, several formulations have been approved by the food and drug administration U.S. (FDA) for clinical use [38,39,40]. Open chain acyclic Gd-chelates when administered have a tendency to leach and expose toxic heavy metals, which are known to induce nephrogenic system fibrosis in patients [41,42], so very stable macrocyclic Gd-chelates are currently under investigation. In this juncture, manganese compounds emerge as a safe alternative *T*1 contrast agent. It does not induce renal toxicity like Gd^3+^; hence, numerous Mn-based NP have been developed as preclinical in vivo *T*1 agents [43,44,45] (Table 2).

Hyeon’s group first reported the synthesis of water-dispersible and biocompatible uniform-sized MnO NPs by thermal decomposition of a Mn-oleate complex and encapsulated in a polyethylene glycol (PEG)-phospholipid shell to render biocompatibility. Relaxometric properties of MnO NPs of various sizes exhibited a trend in which the smaller the particle size, the brighter the signal in *T*1-weighed MR images. This implies that the *T*1 shortening effect increases as the size of the NP decreases. For targeted in vivo imaging of epidermal growth factor receptor (EGFR)-overexpressing cancer cells, Herceptin-conjugated MnO was injected i.v. into a mouse-bearing breast cancer brain metastatic tumor. 

The results showed that breast cancer cells were selectively enhanced in *T*1-weighed MRI for an extended period of time compared to bare MnO NP [47]. Mi et al. developed an activatable magnetic NP for the noninvasive MR imaging of hypoxia and metastasis (Figure 2). The NP is a pH-responsive MRI agent made of Mn^2+^ loaded calcium phosphate (CaP) NP comprising a PEG shell. At a low pH, such as in solid tumors, the CaP disintegrates and releases Mn^2+^ ions. Binding to proteins increases the relaxivity of Mn^2+^ and enhances contrast. In vivo, NP selectively accumulated in tumors with a strong *T*1 signal and is able to identify tumor hypoxic regions and to detect invisible millimeter-sized metastatic tumors in the liver [48]. Lee et al. synthesized a pH responsive *T*1 contrast agent for MR imaging of tumors. The *T*1 contrast agent is a MnO NP trapped in a stable urchin like Mn_3_O_4_ (MnO@Mn_3_O_4_), a hollow container prepared by anisotropic etching. In vivo MRI studies were performed in a mouse bearing NIH3T6.7 tumors. I.v. administration of Herceptin-conjugated MnO@Mn_3_O_4_ demonstrated selective accumulation in tumors and subsequent disintegration of MnO NPs in the acidic tumor microenvironment releasing Mn^2+^ ions. As a result of this event, tumor-specific superior *T*1 contrast was achieved and was observed up to 4 h. The authors suggested that this kind of metal oxide-based contrast agent requires toxicology evaluations due to the leached-out metal ions [49]. Similarly, the ability of MnO NPs to quench fluorophores in the vicinity motivated Wang et al. to co-encapsulate MnO NP and coumarin 545T inside the silica shell for pH-sensitive dual optical and MR imaging of cancer cells. In vitro, a folic acid-conjugated silica shell was selectively taken up by folate receptors overexpressing HeLa cells, and quenched fluorescence of coumarin was recovered due to the acidic leaching of MnO NP in endo-lysosomal compartments. Thus, the released Mn^2+^ ions exhibited strong *T*1 contrast enhancement in HeLa cells [50]. Aptamer-conjugated core-shell NP were developed by Yang et al. for targeted MR imaging of tumors. Core-shell NP consists of magnetic Mn_3_O_4_ core and biocompatible silica shell doped with rhodamine for dual optical and MR imaging, and finally, aptamer was conjugated to the NP surface through a PEG spacer (Mn_3_O_4_@SiO_2_(RB)– PEG–Apt). Incubation with HeLa cells demonstrated the selective uptake of Mn_3_O_4_@SiO_2_(RB)– PEG–Apt, confirmed by both optical and MR imaging. In vivo studies were performed in nude mice bearing HeLa tumors. When administered i.v., Mn_3_O_4_@SiO_2_(RB)– PEG–Apt specifically accumulated in tumors and exhibited significant *T*1 signal enhancement after 12 h [51]. Huang et al. synthesized 5-nm-sized MnO NPs conjugated with poly(aspartic acid)-based graft polymer containing PEG and DOPA moieties via catechol-Mn chelation and finally functionalized with cRGD peptide (mPEG&cRGD-g-PAsp@MnO) to actively target tumors. The *r_1_* relaxivity of mPEG&cRGD-g-PAsp@MnO was measured to be 10.2 mM^−1^ S^−1^ using a 3 T MRI scanner and exhibited negligible toxicity when incubated with the A549 cell line. In nude mice bearing A549 tumors, subcutaneously administered mPEG&cRGD-g-PAsp@MnO slowly accumulated in tumors detected by *T*1 signal enhancement in the period of 30–120 min. Owing to the small size of the contrast agent, researchers suggested that the tumor-specific *T*1 contrast enhancement might be due to co-operative effects of both active and passive targeting [52].

#### 2.1.2. T2 Contrast Agents

The most common *T*2 contrast agent is superparamagnetic iron oxide NP (SPION); high magnetization exhibited by iron oxide causes magnetic inhomogeneities, affecting *T*2 relaxation times. Briefly, dipolar interactions between iron oxide magnetic moment and water proton spins decreases the *T*2 relaxation times, leading to negative image contrast [53]. There are several commercially available *T*2 contrast agents in the market namely, ferumoxtron (Sinerem (EU) and Combidex (US)), ferumoxytol (Faraheme (US)), and ferumoxide (Senti-Scint Feridex (US) and Endorem (Britain)). To improve the accumulation of contrast agents in glioma tumors, Sun et al. developed chlorotoxin (CTX) conjugated PEGylated superparamagnetic iron oxide nanoparticles (SPION). Targeted SPION were highly sensitive to 9 L glioma cells and revealed that increased cell uptake resulted in *T*2 signal enhancement compared to the non-targeted counterparts. Tumor-specific accumulation of targeted SPION was observed in athymic nude mice that bore 9L gliosarcoma tumors confirmed by strong *T*2 signal, and an insignificant amount of non-targeted SPION were also accumulated in tumors by nonspecific mechanisms [54].

For instance, small-sized iron oxide NP (IONP) functionalized with complementary azide and alkyne moieties were engineered to undergo copper-free click chemistry when exposed to matrix metalloproteases (MMPs) overexpressed in the tumor microenvironment. After MMP exposure, IONP self-assemble into superparamagnetic nanoclusters (~800 nm) that amplify *T*2 signal contrast. When incubated with MMP-expressing cancer cells, IONP enhanced the *T*2 signal by 160%. I.v. administration of two sets of iron oxide nanomaterials into tumor-bearing mice demonstrated the *T*2 signal enhancing ability of these self-assembling NP [55]. For targeted MR imaging of brain tumors, Shevtsov et al. developed SPION-conjugated recombinant human epidermal growth factor (SPION–EGF). In vitro studies showed that SPION–EGF is safe and exhibited enhanced cellular uptake. Intravenous administration of SPION–EGF conjugates in tumor bearing rats exhibited targeted delivery across the blood–brain barrier and tumor retention of the NP, evidenced by hypotensive *T*2-signals. These results indicated that SPION–EGF accumulation in tumors was more efficient than unconjugated SPIONs [56]. For the MR imaging of breast cancer, Herceptin cross-linked iron oxide NP (CLIO) was developed by Chen et al. In vitro, CLIO exhibited receptor-mediated endocytosis in Herceptin-overexpressing cancer cells. In tumor-bearing mice, CLIO preferentially accumulated in tumors, as evidenced by the darkening of tumors in *T*2 weighed images [57]. Inflammation activatable neutrophils loaded core/shell mesoporous silica nanoparticles (MSN) for imaging and therapy of residual glioma. Drug-loaded magnetic MSN was internalized into neutrophils to target glioma, and magnetic MSN allowed in vivo tracking and diagnosis of residual glioma and therapeutic guidance. In vivo, neutrophils carrying MSN guided by MRI identified inflamed residual glioma, retained to improve the survival rate, and delayed the tumor relapse [58].

### 2.2. Photo Acoustic Imaging (PAI)

PAI is a hybrid noninvasive imaging technique that combines optical and ultrasound imaging with high spatial and temporal resolution. The principle of PAI is described as, when illuminated with pulsed light, the target tissue absorbs energy and undergoes a transient thermoelastic expansion, which generates ultrasound that can be detected by ultrasonic transducers and converted into an image. Since sound waves are less scattered in tissues compared to light waves, PAI offers increased tissue penetration depth and rich optical contrasts [12,59,60]. Owing to these advantages, PAI is emerging as a potential biomedical imaging technique explored for cancer research applications [61,62]. PAI uses both endogenous and exogenous contrast agents, for instance, with endogenous contrast agents (e.g., hemoglobin (Hb), lipids, water, and melanin), it is possible to obtain structural and functional information, such as lipid distribution, Hb concentration, and oxygen saturation. Exogenous contrast agents offer molecular information about biological events with enhanced image contrast [63,64].

#### PAI Contrast Agents

Among Au nanomaterials, gold nanorods (GNR) have gained significant attention as PAI agent due to their tunable NIR absorption and facile synthesis. The cylinder-like morphology of GNR affects absorption band and absorption shift towards the NIR region with increases in the particle length to width ratio. Hence, better PAI performances can be achieved by adjusting the aspect ratio. Other nanomaterials based on gold used for PAI is gold nanocages (AuNCs); these are cubic NPs with a hollow nanostructure with an absorption band ranging from 600–1200 nm and can be an excellent PAI agent for biological applications. Semiconducting NPs that consist of semiconducting polymers and oligomers exhibit high photostabilty, a large absorption coefficient, tunable optical absorption, and a controllable size and can serve as a PAI agent [65].

Robust NIR-absorbing amphiphilic semiconducting nanoparticles (ASO) for dual in vivo PA and fluorescence imaging of tumors were developed by Yin et al. The self-assembled NP exhibited small size, high structural and photostability, low cytotoxicity, and brighter PA signals in vitro. Following i.v. administration, due to their small size, ASO demonstrated passive tumor-accumulating effects with strong PA and fluorescence signal intensities in the tumor with signal-to-background ratios of ∼3 and ∼27 for PA and fluorescence imaging, respectively [66]. Song et al. developed “smart” gold NPs (SAN) that aggregate rapidly in mild acidic medium as tumor specific photoacoustic agents (Figure 3). Due to the phagocytic nature of the cancer cell, SAN preferentially accumulated in HeLa cells with a strong PA signal that was 17 times stronger compared to normal cells. In addition, intratumoral administration of SAN in HeLa tumor-bearing xenograft mice model resulted in a 5.7-fold PA signal increase at the injection site compared to injection at a normal tissue site. The authors suggested that the application of tumor-specific SAN could be extended to PAI-guided drug delivery [67].

Homan et al. followed biocompatible “green” chemical methods to synthesize Ag nanoplates (Ag NPL) and utilized them as a PAI agent for the targeted imaging of pancreatic tumors. After conjugation with a-EGFR (a monoclonal antibody derivative), Ag NPLs were readily taken up by EGFR-overexpressing pancreatic cancer cells by receptor-mediated endocytosis and exhibited little or no toxicity. Furthermore, in an orthotopic pancreatic mouse model, the potential application of Ag NPL as a targeted in vivo PAI agent was demonstrated [68]. Jokerst et al. proposed a dual PAI/Raman imaging using GNR as a passively targeted molecular imaging agent. GNR with an aspect ratio of 3.5 was selected to image subcutaneous xenografts of various ovarian cancer cell lines in living mice due to their highest ex vivo and in vivo PA signal. A maximum PA signal was observed within 3 h for cell lines tested and lasted up to 2 d post-administration. GNR allowed presurgical PA visualization of a tumor for staging as well as intraoperative SERS imaging for complete resection of tumor margins [69]. In a study, researchers demonstrated the ability of spectroscopic PAI to monitor the passive accumulation of silica-coated GNR in a xenografted tumor as well as its capability of visualizing the presence of exogenous contrast agents in vivo through obtaining PA signals at multiple wavelengths and found a map of a different optical absorber [70]. Similarly, AuNCs were also used as a contrast agent for PAI to detect B16 melanomas at 778 nm and U87MG brain tumors. Kim et al. developed AuNCs with enhanced detection limit as a contrast agent for targeted PA of melanomas in vivo. When conjugated with melanocyte-stimulating hormone, AuNCs served as a novel contrast agent for in vivo molecular photoacoustic tomography (PAT) of melanomas with high sensitivity and high specificity to melanoma, and PA signal enhancement was 300% higher than non-targeted PEG-AuNCs [71]. A dextran-based pH-sensitive NIR nanoprobe was developed and used as a contrast agent for photoacoustic imaging to identify a tumor in vivo. The nanoprobe with pH-sensitive dual NIR resonance absorption exhibited strong pH-dependent PA signals. In a breast tumor bearing mouse model, nanoprobes accumulated selectively in tumor, confirmed by dual-wavelength photoacoustic imaging compared to peripheral normal tissues [72]. Recently, a caspase-3 activatable PA probe for molecular imaging of tumor apoptosis in vivo was developed. Here in, a PA probe recognised by caspase-3 was cleaved to promote macrocyclization reaction followed by self-assembly that leads to tumor-specific PA signal amplification. The prolonged retention of self-assembled probes in tumors enabled the visualization of the activity and distribution of caspse-3 in the entire tumor after doxorubicin (DOX) administration [73]. With a similar strategy using enzyme activatable probes, Wu et al. developed a PA probe for imaging alkaline phosphatase (AP) activity in vivo. In the presence of AP, PA probe undergoes dephosphorylation, causing hydrophilic to lipophilic transition, which leads to signal enhancement. In vivo PAI in HeLa tumors showed that the probe achieved maximal contrast ratio of 2.3 folds (experimental and control) at 4 h after administration [74].

### 2.3. Optical Imaging (OPI)

Optical imaging techniques are rapid, highly sensitive, and noninvasive. They use the physical interactions of light with tissues, leading to several events, such as absorption, scattering, and emission of light, offering biochemical and morphological information [2,75]. For instance, fluorescence imaging techniques involve the absorption of light energy by tissues and reemission as a longer wavelength fluorescence light. In particular, to image deeper tissues, NIR light has been utilized to minimize autofluorescence generated by endogenous light-absorbing molecules, such as hemoglobin [5,13].

#### OPI Contrast Agents

Quantum dots (QDs) are inorganic NP with excellent optical properties and have been widely used in biomedical imaging, diagnostic, and therapeutic applications. Due to the quantum confinement, QDs exhibit size-dependent optical and electronic properties. QDs presents advantages over conventional organic dyes, such as lower photobleaching and size-dependent photoluminescence emission. Gold NP (Au NP) plays an integral role in the biomedical field due to their ease of synthesis and tunable optical and electronic properties [76]. The unique properties of Au NP arise from their surface plasmon resonance. The surface of AuNP can be easily modified with proteins, peptides, oligonucleotides, and many other compounds while still maintaining their optical properties. The advantages of AuNP over their metallic counterpart are their chemical inertness and reduced toxicity [77].

To bypass high liver retention, Tang et al. developed monodisperse, water-soluble silver sulfide QDs with a distinct characteristic light emission from 500 to 1200 nm and a QDs core diameter between 1.5 and 9 nm. When conjugated with cyclic penta-peptide for targeting tumor-associated integrins, confocal imaging studies showed that QDs were selectively taken-up cancer cells with strong fluorescence signals. When i.v. administered, QDs in different mouse models of cancer demonstrated highly selective accumulation in tumors, which resulted in a high tumor-to-liver uptake ratio, an accomplishment rarely achieved by conventional NPs [78]. Similarly, Li et al. developed CdTe QDs with a high quantum yield (38%) for targeted NIR imaging of gliomas. In a mice-bearing U87 tumor, i.v. administered cRGD-QDs conjugate led to targeted imaging of tumor and its margins followed by successful tumor resection, suggesting its potential as a fluorescent indicator for intraoperative tumor imaging [79].

Lee et al. prepared a protease activatable NIR-based nano-probe for in vivo cancer imaging that consists of Au NPs attached to Cy 5.5 dye through MMPs substrate. The nano-probe is designed in a way that the fluorescence of Cy 5.5 is quenched by closely spaced Au NP and will be turned on with exposure to MMPs. Both in vitro and in vivo, nano-probes demonstrated preferential accumulation in MMPs positive cancer cells and tumor tissues with a strong fluorescence signal. Similar studies were carried out by the same research group using polymeric NP [80]. Targeted fluorescence nanoprobes (FNB) that are ultra-sensitive to acidic, angiogenic tumor microenvironments were developed by Wang et al. FNB is a copolymer that consists of an ultra pH-sensitive core conjugated with a series of fluorophores with a large emission range from green to near-infrared (500–820 nm) (Figure 4). The fluorescence of FNB was strongly quenched by homo FRET and rapidly activated (>300) upon exposure to subtle, physiologically relevant pH transitions. I.v. administered FNB demonstrated tumor-specific imaging in a diverse set of animal tumor models with different cancer types and organ sites [81]. The same research group developed pH-responsive “on/off” nanoprobes that allowed highly sensitive, cancer-specific detection of malignant tumors. In this study, the nano-probe design was simplified by replacing the Cy5.5 dye with clinically approved indocyanine green. The resulting pH-activatable nanoprobe amplified the fluorescence signal in the tumor over the normal tissues that allowed real-time, image-guided resection of tumors and occult nodules (<1 mm^3^) in mouse models, which significantly improved long-term survival after surgery [82].

A pH-responsive fluorescent core-shell NP that performs “sense-act-treat” in response to tumor microenvironment was prepared by Shi et al. The nanodevice is made of Au NC core coated with fluorescein isothiocyanate-doped mesoporous silica shell. Finally, the silica shell is modified with thermosensitive rhodamine B-based copolymer, which endows pH switchable fluorescence. In tumor-bearing mice, the nanodevice exhibited tumor accumulation characteristics, confirmed by the transition of the fluorescence signal from green to red. In addition, the nanodevice was also used for chemotherapy and hyperthermia for treating tumors [83]. Zhao et al. developed an oligopeptide-based self-assembling NP that bore an activatable dye and a dark quencher. Fluorescence of the NP was quenched by the dye-dark quencher mechanism, but in the presence of mild acidic pH, structural integrity of the NP was destroyed and the fluorescence signal was recovered. This NP was tested as tumor microenvironment activatable probes for both intratumoral and i.v. in vivo tumor imaging. The authors confirmed that based on distinct fluorescence intensities, the acidic tumor microenvironment can activate stronger fluorescence signals [84]. Li et al. reported a dextran-based NIR fluorescent nanoprobe for noninvasive visualization of tumors in vivo by sensitizing the tumor acidic microenvironment. The nanoprobe in this study is a dextran/polylysine copolymer that bore IR783 fluorophore via the pH labile hydrazone bonds. Fluorescence of the nanoprobe is self-quenched due to closely spaced IR783 molecules in the dextran backbone and led to a low signal background in normal tissues. In the tumor acidic microenvironment, hydrazone bonds are cleaved, which resulted in the NIR fluorescence recovery. In vivo dynamical optical imaging in U87MG tumor xenograft bearing mice showed that the fluorescence of NP increased with time after i.v. administration, and the T/N ratio reached its maximum value of 4.2 at 48 h postinjection [85]. Organic fluorophore with NIR II emission was successfully loaded into amphiphilic matrix for bioimaging in vivo. Nanoprobe allowed noninvasive imaging of blood flow in mice brain; in addition, confocal imaging of fixed biological tissues in a one-photon imaging depth of ~1 mm with sub-10 μm high spatial resolution was also achieved. In vivo two color imaging in NIR II wavelength was performed using the nanoprobe emission between 1100 and 1300 nm and single-walled carbon nanotubes (CNT) emitting above 1500 nm [86].

### 2.4. Nuclear Imaging (NI)

NI is an ultrahigh sensitive, depth-independent imaging technique that has the ability to image any part of the body, including bones, soft tissues, etc. NI techniques available in clinical settings are PET, SPECT, and CT. These imaging techniques use contrast agents that can be imaged by radioactive decay or their ability to attenuate high energy radiation to generate images [14,15,87]. In PET imaging, for instance, radionucleids undergo high positron β^+^ emission decay followed by their immediate annihilation with electrons, generating two γ photons in the process that can be imaged.

#### 2.4.1. Nuclear Imaging Contrast Agents

[^18^F]FDG is the primary agent used for the detection and staging of many cancers using PET. [^18^F]FDG targets tumors with a high metabolism rate and are used in 90% of human PET studies [19]. ^64^Cu is a promising radionuclide with suitable decay (half-life = 12.7 h) characteristics and is known for the convenience with which it can be used for radiolabelling. In addition, ^64^Cu can be readily prepared with high activity by biomedical cyclotron. For SPECT imaging, ^99m^Tc radionuclide is a metastable isomer of ^99^Tc, which is widely used for diagnostic imaging with a half-life of 6.0 h. ^111^In is also a SPECT imaging agent useful for isotopic labelling of biomolecules for specialized diagnostic applications [88,89].

Due to their large X-ray absorption coefficient, Au NP has been demonstrated as an effective CT contrast agent compared to traditional iodine-based contrast agents. Gold exhibits a much higher X-ray absorption coefficient than iodine at 100 keV [90]. Surface modifications of Au NP can enhance absorption coefficients, for instance, the attenuation coefficient achieved by PEG-coated GNPs, to be 5.7 times higher than current iodine-based CT contrast agents [91]. Bismuth (Bi)-based NP as CT contrast agent presents advantages over conventional iodinated and Au-based NP contrast agents due to their high atomic number and to their X-ray attenuation coefficient, low cost, and low toxicity. Bi sulfide holds great promise as a NP contrast agent due to its high effective nuclear charge, physical density, and electron density, which makes them an excellent candidate for CT imaging [92].

#### 2.4.2. CT Imaging

Renal clearable ultrasmall Bi subcarbonate nanoclusters were prepared for tumor-specific CT imaging. These nanoclusters can be assembled to Bi subcarbonate nanotubes (BNTs); this unique structure allowed tumor-specific accumulation followed by disassembly under acidic pH. In addition, DOX-loaded BNTs exhibited excellent therapeutic efficacy when combined with radiotherapy (RT) [93]. Activity-based probes specific to cathepsins were attached to iodinated polymeric dendrimers for the detection of solid tumors using CT imaging. Tumor-specific accumulation of probes enabled CT imaging by activity-dependent covalent binding. In addition, signal detection was achieved using a low dose of 20 mg I kg^−1^ compared to a clinical dose of iodinated agents (300 mg I kg^−1^) [94]. Albayedh et al. studied imaging contrast enhancements in RT by finding relationship between imaging contrast ratio and different parameters such as various NP (gold, iodine, silver, iron oxide, and platinum) concentrations, different beam energies for the different NP concentrations, various beam energies for Au NP, and different thicknesses of the incident layer of the phantom including variety of gold NP concentration. Monte Carlo simulation results showed that Au NP had the highest imaging contrast ratio. In addition, it was proved that a higher contrast will be obtained with high concentrations of NP, low beam energy, and small thickness of the tumor [95].

Shi et al. developed a hypoxia-sensitive nanoprobe that consists of Au NP and nitroimidazole derivative for CT imaging of tumors. After in vivo administration, the nanoprobe exhibited high specificity and sensitivity to image intratumoral hypoxic level based on variations of CT values [96]. Dou et al. prepared a series of Au NP to improve both CT contrast and radiosensitization simultaneously for theranostic application of tumors. AuNP with a size of ∼13 nm demonstrated high CT contrast ability, and significant radiosensitization was used for in vivo studies. In a tumor model, Au NP demonstrated prolonged circulation and tumor-homing abilities compared to clinically used small-molecule agents. The authors suggested that ∼13 nm Au NPs can be utilized as potential clinical X-ray theranostics [97]. A glutathione-induced aggregation of Fe_3_O_4_ NP as a tumor-responsive bimodal imaging probe was developed by Gao et al. The NP prepared in this study is a PEGylated Fe_3_O_4_ that consists of RGD peptide and a self-peptide linked through a disulfide bond. Both in vitro and in vivo, the cleavage of disulfide bonds induced aggregation, which improved MR signal contrast. In tumor-bearing mice, prolonged retention of cross-linked ^99m^Tc labeled NPs enabled enhanced dual MR and SPECT imaging of tumors in vivo [98]. A dual radiolabeled Au NP functionalized with an MMP 9-cleavable peptide was utilized for targeted SPECT imaging of tumors. In vitro, incubation of ^64^Cu labeled Au NP with MMP-9 resulted in the cleavage of the peptide and distinct tumor accumulation properties of the contrast agent were observed between tumors with differing MMP-9 expression [99]. Cheng et al. developed NSC-bearing radiolabeled NPs for targeted SPECT imaging of brain tumors. In this study, MSN used as a nanoplatform was radiolabeled with ^111^In and functionalized with NSC. In vivo SPECT imaging demonstrated the migratory ability of NSC-labeled NPs towards glioma xenografts and enabled visualization of NSC distribution in the brain after their intracranial and systemic administrations [100]. Chen et al. developed core-shell NPs composed of a copper sulfide core and biocompatible mesoporous silica shell and labeled with TRC 105 for the targeted PET imaging of tumor vasculature (Figure 5). In vivo biodistribution studies showed time-dependent gradual accumulation of NP in tumors and reached its peak (6.0 ± 0.4% ID g^−1^) at 24 h after administration compared to non-targeted NP. In addition, the NP was also used for photothermal ablation of tumors [101].

Sun et al. doped ^64^Cu onto core-shell CdSe/ZnS QDs through a cation exchange reaction for PET imaging of brain tumors. In a U87MG glioblastoma xenograft model, PET imaging of radiolabeled QDs demonstrated high tumor targeting ability and reached its peak (12.7% ID g^−1^) at 17 h. Due to its robust nature, the signal of ^64^Cu precisely reflected the biodistribution of QDs, which is confirmed by ex vivo imaging [102]. Zhao et al. developed Au NPs through the incorporation of ^64^Cu into their structures for cancer-specific PET imaging. In a breast cancer mouse model, the alloy NP demonstrated improved pharmacokinetics and a high tumor-targeting ability with an increased T/M ratio confirmed by PET imaging [103]. Liu et al. developed a tumor-specific multimodal theranostic agent based on self-assembled MoS_2_ nanosheets and iron oxide nanocomposites. In a tumor model in vivo, double-PEGylated nanoconstructs demonstrated time-dependent tumor accumulation confirmed by quantitative PET imaging. In addition, in vivo PTT achieved tumor ablation [104]. The tumor-tropic migratory ability of neural stem cells (NSC) was exploited for developing targeted imaging and therapeutic formulations for tumors. Garrigue et al. developed an amphilic dendrimer-based nanoplatform for PET imaging that bore PET reporting units at the terminal. The dendrimer used in this study can self-assemble into micellar structures that can preferentially accumulate in tumors due to the EPR effect for PET imaging with superior imaging sensitivity and specificity. Owing to this unique design, the nanoplatform demonstrated up to 14-fold increased PET signal ratios compared with the clinical PET agent. Importantly, the nano PET agent can image refractory low glucose uptake tumors compared to [^18^F]FDG [105].

## 3. Going Deeper

Imaging technologies have become an indispensable tool in cancer research. At present, a variety of technologies, including optical fluorescence, ultrasonic, CT, MRI, and PET have been used for preoperative staging and have significantly affected how patients with cancer are treated and monitored. However, all of these technologies possess inherent limitations, including poor sensitivity, poor resolution, shallow imaging depth, high cost, and safety issues. We start this section with a discussion of the results from recent literature on the penetration ability of in vivo imaging modalities.

### 3.1. Depth-Dependent

For clinical diagnostics tools, an input signal must traverse the body’s tissues to excite a contrast agent (signal in) and follow it by a reporting signal that traverses the tissues until it encounters a measuring device that can quantify and orientate. Light in the visible light range of 350–740 nm does not penetrate deep into tissue, but NIR wavelengths ranging from 750 to 1000 nm penetrate up to centimeters [5]. In addition, light scattering, autofluorescence, and absorption by adjacent tissues, water, and lipids from in vivo systems constitute fundamental limitations. Therefore, optical imaging is more suitable in superficial (e.g., skin) or other optically accessible (e.g., colon and esophagus) sites for clinical use. So far, some advanced techniques have improved on these limitations [106], and many commonly used animal models, such as zebra fish, mouse, rat, and rabbit are suitable for preclinical research of fluorescent imaging agents and photo-based therapy. Green fluorescent protein has been utilized extensively to study biological processes, such as gene expression and cell localization [107,108]. In our previous study, HB1.F3.CD cell expressing firefly luciferase were used to trace the spatial distribution of neural stem cells. Bioluminescence imaging was performed using an in vivo imaging instrument and demonstrated the delivery of MSN-DOX via tumor tropic HB1.F3.CD carrier cells over a distance of 5 mm [109]. One of the key strategies for imaging deeper tissues inside samples has been the use of NIR light to minimize tissue autofluorescence and to further improve target/background ratios. NIR dye offers advantages over visible ones due to the absorption and scattering properties. Weissleder and Ntziachristos experimentally showed fluorescence imaging through the body of a nude mouse at 532 nm and 670 nm and demonstrated that the signal was four-orders of magnitude stronger for illumination in the NIR than illumination with green light under otherwise identical conditions. Furthermore, fluorescent dyes that emit in the NIR spectrum, which allow identification of deeper tumors, were utilized to study orthotopic C6 glioma tumor by Ruan et al. [110]. The overexpression of legumain in a glioma site led to the selective accumulation of AuNP-Cy5.5-A&C, and fluorescence signal of AuNP-Cy5.5-A&C began emerging in the glioma site with an increasing intensity over time and obtained a high G/B ratio (the ratio of fluorescence at the glioma site to fluorescence at the normal brain site). A drawback of most fluorescence agents is that they are subjected to strong scattering and absorption at tissue, which not only causes the rapid loss of its intensity but also reduces contrast in confocal images when focusing more than a few tens of micrometers into tissue [111,112]. Multi-photon luminescence refers to a fluorophore can be excited within a time window (between atto- and femto- seconds, 10^−18^–10^−15^ s) by two or three photons that have a half or a third of the energy required to fill the gap between two of its energy levels [113]. For this reason, IR light with a high penetration ability is required for multiphoton excitation instead of UV or visible light. A specific example of using an extrinsic fluorescent label and intravital microscopy for NP tracking was performed by Cheng and coworkers [114]. In that study, individual metabolic pathways of charged MSNs labeled with the fluorescent molecule fluorescein isothiocyanate were observed using two photon microscopy. The dynamic information acquired serially at 4 s intervals revealed that hepatocytes showed preference for positive-charged MSNs and that negative-charged ones were rapidly taken up by Kupffer cells. Two-photon (TP) excitation can be used not only for enhanced depth of imaging but also to achieve efficient PDT of deep-seated tumors [115,116,117]. A key feature of TP excitation is the non-linearity of photon absorption that makes it possible to activate PSsat the focal point. This allows precise three-dimensional (3-D) manipulation during PDT for some specific diseases in the eye or brain, reducing off-target damage to surrounding healthy tissue. However, conventional PS, such as porphyrin and metalloporphyrins, which have low two photon absorption (TPA) cross sections, greatly hamper TP-PDT. To address this issue, some NPs, including semiconductor QDs [118,119], silica-based NPs [120,121], Au NPs [122], and polymeric nanomaterials [123], were designed as TPA materials to sensitize encapsulated PS molecules. In our previous study, we designed a well-ordered mesoporous structure of MSNs that co-encapsulate TPA dyes. The unprecedented 93% of interface energy transfer efficiency was observed in 3-D hexagonal MSN structures and demonstrated the cytotoxicity induced by the singlet oxygen-generated in vivo breast cancer models.

In contrast to the “light in-light out” modality of optical imaging, PAI detects a reporting signal by means of ultrasonic emission as a result of heat-induced transient thermoelastic expansion (light-in sound-out) [124]. Since the conversion from optical to ultrasonic energy reduces the optical diffusion limit by capitalizing on the low acoustic scattering in tissue, it could overcome the limitations of tissue penetration to permit detection up to multi-centimeters deep into tissues compared to optical coherence or microscopy techniques alone, without completely sacrificing resolution [125]. In other words, PAI combines the advantages of both optical imaging and ultrasonography for tissue imaging. In addition, unlike optical imaging that requires additive contrast (reporter) agents, some endogenous substances with special light absorption, such as Hb [126], melanin [127] and lipid [128], have been used as PAI contrast agents for biomedical applications. For example, Hb, an iron-containing metalloprotein in red blood cells, is essential to tissue metabolism. It carries oxygen throughout the body to permit aerobic respiration to provide energy to power the functions of the organism. Because oxy- and deoxy-Hb each possess different light absorption spectra that can be employed to generate a unique optoacoustic signal, images of each endogenous contrast agents can then be reconstructed in analogy to the formation of ultrasound images by using appropriate mathematical methods. In particular, multispectral optoacoustic tomography employs multiple optical wavelengths and spectral unmixing algorithms, which provides the ability to recognize specific absorbers. For example, Mallidi et al. reported that PAI could provide a 3-D atlas of tumor saturation O_2_ concentration (StO_2_) by measuring oxy- and deoxy-Hb to predict PDT treatment efficacy. As shown in their study, the drug–light interval that responds to PS location influenced treatment efficacy. More interesting, the 3-D tumor StO_2_ maps can be used to predict tumor regrowth within 24 h of treatment, thus making early intervention possible [129].

In living subjects, despite endogenous chromophores, such as melanin or oxy- and deoxy-Hb that maybe utilized to form PA images, other intrinsic or extrinsically administered NP would contribute to an undesirable background. Consequently, exogenous PAI contrast agents with strong NIR absorbance and high photothermal conversion efficiency are important for molecular imaging to provide high sensitivity, specificity, and signal-to-background in deep tissue. Metal NPs, especially Au NP, have been widely used as PAI contrast agents in recent years. Zhang and coworkers’ study demonstrated for the first time that AuNPs may be used as a PAI contrast agent for in vivo tumor imaging [130]. However, the optical absorption spectrum of Au NP is similar to that of blood, which makes it difficult to distinguish individual particles from blood in vivo. To address this challenge, many complicated gold nanostructures with tunable surface plasmon resonance peaks in the NIR region have been introduced as PAI contrast agents [131,132,133,134,135]. Very recently, PAI has been shown to detect cancer in clinical applications [136,137,138,139,140]. It demonstrated improved sensitivity and specificity compared with the current gold standard technique and is considered a possible replacement diagnostic strategy in medical diagnostics. Core/shell QDs that consist of a CdS shell and a lead sulfide (PbS) core were developed for NIR II imaging at the 1600 nm wavelength. The fluorescent properties of PbS core were retained by a chemically inert CdS shell after phase transfer to an aqueous environment to reduce toxicity. PEG-CdS/PbS QDs demonstrated better tumor accumulation properties due to prolonged in vivo half-life. As a result, a high tumor-to-normal tissue ratio up to ~32 was achieved and able to visualize mice tumor vasculatures in vivo with an imaging depth of ~1.2 mm [141]. A novel peptide-tagged NIR II fluorescent probe that actively binds to tumor stem cell biomarker was developed for targeted tumor imaging. A targeted probe demonstrated efficient tumor accumulation with a tumor-to-normal tissue ratio of 8. Compared to conventional peptide-dye probes, the probe developed in this study undergoes rapid renal excretion at a rate of 87% in 6 h and allowed fluorescent imaging of the urethra through intact tissue [142].

### 3.2. Depth-Independent

While optical imaging offers excellent sensitivity, good temporal resolution, and multiplexed capabilities, it suffers from limitations regarding depth issues compared with the limitless depths achievable with CT, MRI, and radionuclide techniques. Among them, MRI as a nonionizing imaging modality offers high temporal and spatial resolution (clinical: ~1 mm compared with 5–7 mm for PET, preclinical: micrometers, as opposed to millimeter resolution achievable via optical and radionuclide imaging; Table 1) and superb soft tissue contrast. For example, in recent years, Gd^3+^ chelate contrast agent enhanced MRI is the primary method for detection and preoperative localization of brain tumors to provide anatomical structure for therapy. In comparison to conventional Gd^3+^ chelates, Sun et al. designed an MRI nanoprobe that targets matrix MMP-2 overexpression gliomas through high selectivity and binding affinity of CTX peptide [54]. Combined with prolonged retention in tumors and active targeting, the targeted contrast enhancement of tumor cell signal was demonstrated with in vivo small animal MR imaging. Apart from providing detailed anatomical information, MRI is capable of providing physiological information via various specialized MRI techniques, including dynamic contrast-enhanced MRI, diffusion-weighted MRI (DW-MRI), and blood oxygen level-dependent MRI [143,144]. Cheng et al. reported using DW-MRI to analyze 78 patients, and the apparent diffusion coefficient (ADC) was calculated and compared in the differentiation between malignant and benign lesions. They showed that the mean ADC values of malignant tumors (1.08 ± 0.16) × 10^-3^ mm^2^ s^−1^ were significantly lower than those of benign solid lesions (1.68 ± 0.33) × 10^−3^ mm^2^ s^−1^ and supposed that DWI can be applied as a complementary tool in the differentiation of benign and malignant lesions [145]. Nevertheless, the poor sensitivity of MRI constitutes a major limitation compared with other molecular imaging modalities. This low sensitivity can lead to relatively long acquisition times, and large amounts of imaging agents are often introduced to obtain an adequate signal. These large amounts of imaging agents (i.e., many log orders higher compared with those needed for PET or SPECT; Table 1) can be problematic due to the likelihood of altering the biological system of interest through pharmacological effects. For this reason, iron-labeled stem cells involve preloading the cells with SPION prior to administration. Subsequently tracking their migration and tumor distribution over time with MRI is a promising method to improve its sensitivity and toxicity [146,147]. The first-in-human FDA-approved investigational use of ferumoxytol to label NSCs was reported in 2013. The MRI revealed that the tropism capacity of HPF-labeled NSC to glioma following intracerebral or i.v. injection is a promising noninvasive method for clinical use.

CT imaging is another depth limitless technology that can image hard tissue that attenuates high-energy radiation [148,149]. Unlike traditional X-ray examinations, CT obtains signals by a large number of detectors at different angles and processes them with a computer that enables CT to produce high-resolution anatomical images. Upon CT, tissues or media with different X-ray attenuation reflecting these differences in density and composition of anatomical information can be quantified to enable efficient difference delineation between various structures. Godfrey N. Hounsfield used Hounsfield units to define the differences in density between various tissue types (e.g., HU for bone = 1,000, blood = 40, cerebrospinal fluid = 15, water = 0, fat = −50 to −100, and air = −1000). In the clinic, radiologists utilize software that automatically assigns HUs to acquire various diagnostic images, including bone, head, lungs, heart, and abnormal tumor areas. High spatial resolution, rapid acquisition rate, relative simplicity, and good availability constitute major advantages of CT use.

As mentioned above, soft-tissue contrast is quite poor. Therefore, to generate soft-tissue contrast, extrinsic contrast agents are required. Iodine-based contrast agents have been widely applied in clinical diagnosis, including extracellular fluid, vascular space, hepatocellular, and tissue-specific imaging [150,151]. Moreover, a number of research groups have recently reported the use of high atomic number NPs, such as Au [152,153], Gd [154,155], Hf [156], and Bi [157,158] in oncology to assess and track changes in tumor neo-vasculature and has enabled clinicians to evaluate tumor response to therapeutic agents [159,160]. Amongst the emerging NP-based contrast agents, AuNP, which exhibits high x-ray attenuation, biocompatibility, facile synthesis, and surface functionalization, has been introduced into commercial products, e.g., Aurimmune and Auroshell [161]. Compared with other imaging modalities, the relatively high mass concentration of contrast agents that is necessary is one of the main limitations of CT (CT typically requires millimolar concentrations, while MRI can detect micromolar concentrations) [153,158]. In addition, overexposure to X-rays may increase the probability of lifetime cancer mortality, which limits the number of scans that can be performed in the same patient in a given period. Moreover, some clinical contrast agents of CT can be problematic (due to renal toxicity) and cannot be given to all patients, especially in elderly patients with high creatinine levels or pregnant women. Nanobody libraries were prepared for the targeted immune PET/CT imaging of extracellular matrix proteins overexpressed in cancers and other diseases. For instance, NJB2 nanobody that actively binds to fibronectin (an ECM protein) overexpressed in tumors, metastases, and other diseases was validated. ^64^Cu-labeled NJB2 was able to readily detect primary and metastases in various cancer models. Importantly, excellent sensitivity of the nanobody allowed for the precise imaging of breast cancer metastases and early lesions of pancreatic cancer, which are otherwise difficult to image clearly with conventional clinical PET agents [162]. In another study, PET was used to image two important factors of glioma growth, inflammation, and proliferation, namely translocator protein and MMPs. Herein, two PET radiotracers [^18^F]DPA-714 and [^18^F]BR-351 were used to assess the protein localization and quantification of expression in the pathogenesis of glioma. The results revealed distinct PET signals corresponding to the areas of glioma activity within the heterogeneous glioma tissue [163].

Compared to CT, both PET and SPECT are ultrahigh sensitive radioactive imaging techniques that enable evaluation of function and levels of biomolecules within a living subject. In PET imaging, a pair of γ photons, which are generated by radioactive isotope decays, can move in approximately opposite directions and travel much further distances than visible and near-infrared photons in tissue. Therefore, the scintillator can be operated to detect γ-rays and to reconstruct them into tomographic images with good time resolution. Various radioactive isotopes have been used, including ^18^F (t_1/2_ = 109.8 min), ^15^O (t_1/2_ = 2.03 min), ^13^N (t_1/2_ = 10 min), ^11^C (t_1/2_ = 20.3 min), ^14^O (t_1/2_ = 2.04 min), ^64^Gu (t_1/2_ = 12.7 h), ^124^I (t_1/2_ = 100.22 h), ^76^Br (t_1/2_ = 16.2 h), ^82^Rb (t_1/2_ = 1.27 min), and ^68^Ga (t_1/2_ = 68 min). Among them, the ^18^F-labeled imaging agent is the most commonly used in current clinic applications in the areas of cardiology, neurology, and oncology [164,165]. Ido et al. substituted the 2-carbon hydroxyl group of glucose for a radioactive fluorine atom (^18^F) to synthesize the first ^18^F-labeled imaging agent, [^18^F]FDG. Brown and coworkers then demonstrated that [^18^F]FDG was rapidly trapped by tumors through glucose transporter and was resistant to further metabolic processes. Therefore, [^18^F]FDG played a role as an indicator to provide 3-D images of the tumor to be visualized [166]. Recently, Andrade et al. and Grgic et al. used the standardized uptake value (the decay-corrected tumor activity concentration divided by injected activity per unit body weight, surface area, or lean body mass) to predict survival in breast carcinomas and solitary pulmonary nodules, respectively [104,167,168]. In addition to its clinical utility, PET has a wide range of applications in basic research and preclinical arenas. For example, PET can be utilized to investigate basic physiological and molecular mechanisms of human disease through the use of appropriate radiolabeled imaging agents and animal models [104,169,170]. The insufficient spatial resolution and lack of an anatomical reference frame constitute the main disadvantages of PET. This shortcoming has recently been addressed by combining these instruments with either CT or MRI, enabling accurate identification of molecular events with precise correlation to anatomical findings [171,172,173].

SPECT uses nuclides, such as ^99m^Tc (t_1/2_ = 6 h), ^123^I (t_1/2_ = 13.3 h), and ^111^In (t_1/2_ = 2.8 d), which decay via the emission of single γ-rays with differing energies [104,124]. Despite the sensitivity of SPECT being less than PET, based on the isotope-specific energies of the emitted photons, imaging multiple different targets simultaneously (“multiplexing”) with dual or more isotopes is possible with SPECT. Guo et al. took advantage of this characteristic and devised a dual-isotope experiment to assist the diagnosis of hepatic tumor and liver fibrosis. In their study, the tracers of ^131^I-NGA and ^99m^Tc-3P-RGD_2_ were selected to target the asialoglycoprotein receptor on the hepatocytes and the integrin α_v_β_3_ receptor in tumors or fibrotic livers, respectively. Clear fusion images facilitate the acquisition of different physiological information for diagnosing liver fibrosis and liver cancer and for evaluating residual functional liver volume simultaneously [174]. Additionally, since the radionuclides commonly available for SPECT have longer half-life periods (ranging from a few hours to days), SPECT imaging can be used not only for monitoring a site of disease but also for longitudinal tracking. A longitudinal research of lung cancer progression was carried out by Price et al. Their data indicated that radiotracer ^99m^Tc-pertechnetate could be reliably traced in both subcutaneous and orthotopic xenograft tumor models for more than 47 d [175].

## 4. Image Guided Therapy (IGT)

The future of molecular imaging is indeed very promising. The continual development in the field not only provides more precise, rapid, and early detection of tumors but also assists the prediction of tumor therapy and recurrence. Currently, IGT aims to use imaging to improve the localization and targeting of diseased tissue and to monitor and control treatments. In IGT, radiation therapy or surgical planning, advanced anatomical imaging technologies, such as MRI, CT, and PET, provide high spatial resolution to reveal anatomical structures and delineate the targets and relevant normal tissues on those images, which can minimize invasive interventions [176]. For example, Fried et al. were pioneers in MRI-based surgery. They report the first endoscopic surgeries using intraoperative guidance with open MRI for performing endoscopic sinus surgery on 12 patients. The image plane was surgeon-controlled, and the MRI updated images in as little as 14 s [177]. Furthermore, RF ablation that induces thermal injury to the tissue through electromagnetic energy deposition has been one of the major minimum invasive tools in the treatment of liver malignancies. For planning the ablation procedure, targeting of the lesion can be performed with ultrasound, CT, or MRI to evaluate whether any adjacent normal structures are being affected at the same time [176,178]. Attributing the success to imaging guidance, RF ablation appears as the preferred treatment for patients with early-stage hepatocellular carcinoma (HCC), especially in patients with very early HCC in which the complete response rate approaches 97% and with five-year survival rates of 68% [179]. Although considerable research efforts have been invested into minimum intervention of diagnosis and therapy procedure, today’s theranostics still present some major challenges.

So far, present therapy methods, including surgery, chemotherapy, and radiation therapy, can be utilized to treat numerous forms of cancer. However, after all of these treatments, many patients still suffer from unintended side effects or tumor recurrence. For instance, surgery, in many instances, is not able to completely remove all cancer cells. In addition, chemotherapeutic agents and RT damage surrounding healthy tissues, as they are employed to kill cancer cells. Over the past two decades, NPs have emerged as a suitable drug delivery platform for overcoming pharmacokinetic limitations associated with conventional molecular formulations. NP-based contrast agents have attracted growing interest as imaging probes and drug carriers. More recently, advances in theranostics have led to the development of nanocarriers that combine diagnosis drug delivery and therapy monitoring to achieve noninvasive IGT (Figure 6) [180,181].

Phototherapies induced by light enable selectively killing cancer cells under light irradiation through exciting phototherapeutic agents [182,183]. Well-engineered phototherapeutic agents could selectively target tumors, followed by spatially controlled light illumination, to selectively ablate cancer cells and to cause rapid damage fall off outside the target. Such dual-selectivity offered by phototherapies could significantly reduce systemic toxicity associated with traditional chemo- or radio-therapeutic approaches. PTT employs photoabsorbing agents (e.g., liposomes, polymeric NP, magnetic NP, QDs, carbon-based nanomaterials, and MSN) to generate heat from light, leading to thermal ablation of cancer cells and subsequent cell death. Liang and coworkers carried out noninvasive imaging guided PTT in 2014 [184]. A rather low dose (ca. 0.2 mg kg^−1^) single-walled CNT (SWCNT) was utilized as a NIR light-absorbing agent to ablate both the primary tumor and metastasized cancer cells in the lymph node. In this work, NIR optical imaging allowed clear visualization of lymph node tumors in mice with superior imaging resolution and high signal/background ratios. On the other hand, MR imaging provided lower spatial resolution and displayed whole-body imaging.

The use of PAI during PTT offers a major advantage due to simultaneous diagnosis and therapeutic planning. In the study by Huang et al., it was found that the photothermal heating of self-assembly of biodegradable gold vesicles could induce cancer cell death after irradiation and PAI was employed to guide the process [185]. A number of other groups have also reported the use of PAI as a photothermal detector for guiding PTT [186,187,188,189,190].

Light-induced PTT based on photoabsorbing agents with high photothermal conversion efficiency can induce cell apoptosis/death. However, the limitation of light-penetration depth restricts their further in vivo applications and rapid clinical translations in deep-seated tumors. The first depth-limitless hyperthermia treatment was reported in 1957, in which Gilchrist et al. used magnetic particles with alternating magnetic field (AMF) to kill lymphatic metastases by inductive heat [191]. Since then, many studies have followed that harnessed this technology for potential clinical use [192,193,194]. In addition, magnetic NP can potentially achieve theranostics through combining MRI and magnetic hyperthermia treatment. This is because they not only enhance the *r*_2_ relaxivity of protons and lead to a darkening in the reconstructed image but also generate heat by the application of alternating current magnetic fields. Regardless of PTT or AMF hyperthermia, heat loss through convection by means of blood circulation limits the volume of ablation. To minimize heat loss by blood circulation, several strategies for reducing blood flow during ablation therapy have been proposed. For example, Visaria et al. developed PT-cAu-TNF-α, which was composed of 33 nm polyethylene glycol-coated colloidal Au NP and TNF-alpha, to suppress tumor blood flow. Pretreatment with PT-cAu-TNF-α induced vascular damage and significantly reduced tumor survival to 0.005% [195]. A small molecule NIR II fluorophore was conjugated to an amphiphilic peptide for optical image guided PTT. The self-assembled probe exhibited prolonged circulation time in vivo and retained in the tumor visualized by strong fluorescent signals. In tumor models, the probe demonstrated antitumor efficacy by photothermal ablation under a low dose of probe and light irradiation [196].

Different from PTT, which relies on photothermal heating to “cook” cancer, photodynamic therapy (PDT) is based on the interaction between the excited PS and surrounding molecules, generating reactive oxygen species (ROS) to kill cancer cells [197]. According to the mechanism, PS are categorized into either type I or type II reaction. First, PS can directly transfer a proton or an electron to the cell membrane or a molecule and can form a radical anion or cation, respectively, which then reacts with oxygen to produce oxygenated products, such as superoxide anion radicals, hydroxyl radicals, and hydrogen peroxides (type I reaction). Alternatively, the singlet excited-state PS is very unstable and transfers energy to molecular oxygen through a process known as “intersystem crossing”. This energy-transfer step leads to the formation of singlet oxygen (^1^O_2_), and the reaction is referred to as a type II reaction. It is to be noted that both type I and type II reactions can occur simultaneously, and the ratio between these processes depends on the type of PS, as well as the concentrations of molecular oxygen and substrate present. However, most of the studies indicate type II reactions, and thus, ^1^O_2_ plays a dominant role in PDT [198,199,200].

PDT is an externally activatable treatment modality for various diseases and has already been approved for cancer treatment clinical settings because it is a less invasive technique and enables control of the area of photodynamic effect, leaving the surrounding healthy tissues undamaged. Upon administration of PS molecules, the lesion is then selectively illuminated with light of an appropriate wavelength, which, in the presence of oxygen, leads to the generation of cytotoxic oxygen species (e.g., ^1^O_2_ and free radicals) by PS molecules and consequently to cell death and tissue destruction. Compared with conventional PDT, the emerging NP-based PDT could increase PS payload, could effectively target cancer and maximize tumor damage, and could thus improve the therapeutic efficacy and specificity of PDT. In addition, nanotechnology provides a platform for the integration of multiple functionalities in a single formulation that allows imaging to guide the process of therapy. Ling and coworkers used a newly developed NP delivery system consisting of Ce6 grafted poly(ethylene glycol)−poly(β-benzyl-L-aspartate) (PEG-PBLA-Ce6) with an incorporated extremely small iron oxide NPs payload. Such pH-sensitive magnetic nanogrenades (PMNs) remained in an “off” state with no phototoxicity with light exposure due to the FRET-based self-quenching effect of PS within the NP core. However, *T*_1_ MR imaging and the photoactivity of PMNs can be turned on when present in the tumor pH environment. Their results indicated that the synthesized multifunctional PMNs were effective for enhanced PDT of tumors in vivo and reached very early-stage diagnosis. In a clinical study, oral application of 5-ALA leads to highly specific accumulation in malignant glioma tissue that assisted fluorescence-guided resection and implemented intraoperative PDT. Many studies reported extended median survival time for photodynamic diagnosis/intraoperative PDT groups as compared to the control group [201,202,203]. Unfortunately, most of the PS can only be activated by visible light which cannot pass through thick tissue. Recently, new PS have been developed to shift their maximum absorption to the NIR region. However, for practical applications, NIR light can still only penetrate 5 mm into the tissue because enough energy needs to be reserved for PS activation.

Aggregation induced emission (AIE) of several fluorogens are increasingly recognized in biomedical applications. Such compounds exhibit little or no emission in molecular state but, when induced, begin to emit in the aggregate state due to the restriction of intramolecular motions and prohibition of non-radiative decay [204,205]. Based on this idea, tumor-specific rapid imaging was achieved using a bio-orthogonal fluorescence turn-on probe. Using an alkyne group containing a fluorogen probe upon bio-orthogonal reaction with expressed azide on tumor cells, fluorescence was significantly enhanced, enabling tumor selective imaging. In addition, guided by the turn-on fluorescence signal, antitumor efficacy was realized by PDT using the probe as a PS [206]. Likewise, a targeted theranostic AIE nanodot for image-guided PDT was developed. The obtained nanodots exhibited red fluorescence and generated ^1^O_2_ with high efficiency in aggregate state. In in vivo tumor models, nanodots exhibited tumor-targeted imaging, which enabled image-guided PDT for tumor ablation [207].

The in vivo assembly of NIR II emitting down conversion NP decorated with DNA and targeting moieties was developed for image-guided surgery of metastatic ovarian cancer. Nanoprobes demonstrated excellent targeting and tumor retention properties. NIR II fluorescence of assembled nanoprobes was able to delineate tumor margins and tumors that enabled resection. Due to superior tumor to normal tissue ratio, metastases with ≤1 mm were completely excised under NIR-II imaging guidance [208].

Upconversion NPs (UCNP), which emit high-energy visible light under NIR excitation, exhibit improved tissue penetration depth, as compared to traditional down-conversion fluorescence. Their higher photochemical stability and no autofluorescence background make new NIR-induced PDT widely explored. Different from TP-excited PDT, which requires a pulsed laser as the excitation light, UCNP-based PDT can be excited by continuous lasers with much lower instant energy densities [209,210]. The first in vivo UCNP-based PDT study in animal experiment was performed by Wang et al. In their study, a water-soluble UCNP-Ce6 nanocomplex displayed an obvious tumor regression effect on 4T1 tumor-bearing BABL/C mice after tumors were exposed to 980 nm light at 0.5 W cm^−2^ for 30 min [211]. Soon afterwards, an imaging-guided UCNP-based PDT study was also conducted by Wang and coworkers. They utilized the intrinsic optical and paramagnetic properties of Mn^2+^-doped UCNP for in vivo dual modal imaging and observed a significantly increased tumor accumulation of pH-responsive charge-reversible UCNP-DMMA-PEG after i.v. injection [212].

UCNP, with its unique upconversion optical properties, have significant potential in image guided-photodynamic therapy (IG-PDT). However, the low quantum yield of upconversion luminescence (UCL) emission of UCNP (less than 1% for most of UCNP) has been an important issue that reduces ^1^O_2_ generation efficiency. In addition, most PS are type II reactions, and PDT are oxygen-consuming modalities. The therapeutic effects of PDT will be severely affected in the absence of tissue oxygen. To address this issue, Hou et al. developed a new upconversion-based TiO_2_ photosensitizing nanoplatform that generated ROS by means of an electron–hole pair that catalyzed further redox reactions [213]. Under 980-nm irradiation, core/shell UCNP can provide enhanced upconverting UV emission with an energy greater than the band gap energy of TiO_2_. The electrons in the valence band of TiO_2_ are excited to the conduction band, thus resulting in the formation of photo-induced hole−electron pairs which possess strong reduction and oxidation properties that can interact with surrounding O_2_ and H_2_O molecules to generate various ROS [214,215].

Although PDT has the potential for the treatment and management of cancers, it is currently only being used in the treatment of superficial lesions or lesions that are accessible through endoscopes. The fundamental problem lies in the inability of PDT to treat solid, bulky, or deep-seated tumors. Very recently, a few reports have explored some revolutionary PDT strategies, which employed depth-limitless X-ray to activate PS (Figure 7) (Table 3) [216,217,218]. Scintillation NP, such as LaF_3_:Tb^3+^, SrAl_2_O_4_:Eu^2+^, LaF_3_:Ce^3+^, Tb_2_O_3_, LiYF_4_@SiO_2_@ZnO, and GdEuC1_2_, could emit visible luminescence upon exposure to ionizing irradiation, such as X-ray, which shows remarkably better tissue penetration ability as compared to visible and NIR light [219,220,221,222,223,224]. Thus, lanthanide-doped fluoride/oxide NP, such as LaF_3_:Tb^3+^/Ce^3+^ or Tb_2_O_3_, are often utilized as X-ray energy acceptors. Bulin et al. reported that Tb_2_O_3_@SiO_2_ porphyrin NP could be activated by X-ray irradiation with the energy transferred from Tb_2_O_3_ NP to porphyrin to generate ROS. The very good overlap between the emission peaks of Tb_2_O_3_ NP and the absorption peaks of porphyrin led to efficient energy transfer [219]. In another study, Ce^3+^-doped lanthanum(III) fluoride (LaF_3_:Ce^3+^) NP with strong green emission at approximately 520 nm overlapped the absorption peaks of PpIX. Upon X-ray irradiation, energy transfers from LaF_3_:Ce^3+^ NPs to PPIX resulted in ^1^O_2_ generation and the killing of cancer cells due to oxidative stress, mitochondrial damage, and DNA fragmentation [225]. The highlight of this approach is that there is no limitation to the penetration depth achievable by X-rays in tissue. Chen et al. reported a novel, SrAl_2_O_4_:Eu^2+^-mediated X-ray inducible PDT (X-PDT) methodology. An in vitro cytotoxicity study supported that X-PDT induced significant cell apoptosis, even with a 4.5-cm thick pork positioned between the X-ray source and cells [223]. In another study by the same group, X-PDT combined with RT was employed to eradicate H1299 tumors that were either subcutaneously inoculated or implanted into the lung of mice [226]. Recently, a nanoscintillator based on Gd_2_(WO_4_)_3_:Tb was developed and evaluated its potential as dual (MRI and CT) mode imaging agent and combined X-ray and X-PDT agent. Bimodal imaging ability arises due to Gd as a *T*1 contrast agent for MR imaging and a high Z element W serves as a CT contrast agent. In the presence of X-ray, the energy transferred from the GWO matrix to Tb emits green fluorescence, which could effectively activate a PS for X-PDT, while W serves as a radiosensitizer for RT. Guided by MR/CT imaging, synergistic therapeutic effects of nanoscintillators were realized in vivo tumor models [227].

An alternative approach involving activating PS associated with Cerenkov radiation (CR) has been demonstrated by Achilefu et al. [^18^F]FDG carried out β emission decay to generate CR, which was used as an internal light source to activate TiO_2_ NP to produce free radicals without requiring oxygen. In this way, [^18^F]FDG could not only serve as a PET imaging isotope but also acquire the function of providing light for phototherapy. When a single dose (1 mg kg^−1^) was injected into the bloodstream with [^18^F]FDG, TiO_2_−Tf−Tc was found to have the most significant treatment effect and extended survival by more than 20 d [229].

Finally, X-rays with energy in the megavoltage range are commonly used for conventional RT. However, few studies have reported whether the scintillation NPs could be excited by megavolatage X-rays still improve the energy transfer for the significant killing of cancer cells. To address these critical issues, Clement et al. estimated that the singlet oxygen dose generated from CeF_3_-verteporfin conjugates were (1.2 ± 0.7)  ×  10^8^ and (2.0 ± 0.1) × 10^9^ singlet oxygen molecules per cell for a therapeutic dose of 60 Gy of ionizing radiation at energies of 6 MeV and 30 keV, respectively. A significant Panc 1 pancreatic cancer cell-killing efficacy was reported from the combined radiation and PDT treatment when compared to control groups [228]. These results could be highly useful for clinical applications in the prediction of light dosimetry. Berbeco et al. developed a trimodal theranostic NP (SiBiGdNP) for the amplification of image contrast and clinical radiation dose. When administered in a non-small cell lung carcinoma model, the theranostic NPs demonstrated that simultaneous MR, CT contrast enhancement, and X ray irradiation (10 Gy) produced statistically significant improvements in both tumor growth delay and survival compared to the control group. A targeted upconversion NP (arginine-glycine-aspartic acid-labelled BaYbF_5_:2%Er31) was developed by Shi et al. for CT imaging and CT imaging-guided RT. The theranostic NP showed low toxicity and in vivo RT in the presence of NP demonstrated shrinkage in tumor volume percentage of 35%, with growth inhibition that lasted for at least 12 d. Wu et al. developed gadolinium-doped carbon dots theranostic NP (TNP) for MRI-guided RT. TNP (~18 nm) was biocompatible and exhibited prolonged circulation time (6 h). In a tumor xenograft model, TNP showed a radiation dose enhancement effect with a significant shrinkage in the tumor volume of 53% by radiosensitized RT compared with RT alone. Zhao et al. developed multifunctional tungsten sulfide QDs (WS_2_ QDs) for the dual modal image guided PTT and RT. In a mice-bearing tumor, intratumoral administration of WS_2_ QDs (3 nm) significantly enhanced CT and PAI contrast and also exhibited significant inhibition in tumor growth due to the combined PTT and RT. In addition, H and E staining analysis confirmed that WS_2_ QDs is safe with no observable toxicity.

## 5. Conclusions

The application of NP in biomedical fields arises mainly to address critical limitations of imaging or chemotherapeutic agents. NP can 1) significantly increase the aqueous solubility of poorly water-soluble therapeutic agents many fold compared to their bare analogues; 2) prolong their in vivo circulation times, which reduces the frequency of administration; and 3) reduce their undesired nonspecific distribution, thereby preventing life-threatening side effects and improving the patient’s quality of life. Owing to these beneficial features and vast developments in the chemical and biological sciences, sophisticated NPs with multiplexing capabilities were developed, which earned recognition as clinically relevant potential therapeutic agents for the treatment of cancer. As we discussed in this review how a large number of NP with innovative designs were utilized in the accurate early diagnosis of cancer, the potential information gained, such as spatiotemporal location, size, and shape of tumors, assists clinicians to arrange and plan suitable treatment regimens. In addition, we also discussed various NP-based therapeutic strategies that can eradicate deep-seated tumors either by single treatment or through multimodal cooperative interactions through a combination of treatments. NP offer these exciting possibilities, which can be hardly imagined otherwise.

Although NP holds great promise as therapeutic agents, their clinical translation is still in its infancy. This is largely due to the lack of sufficient knowledge regarding NP pharmacokinetics and underlying physiological processes in biological environments. In addition, evaluation of the biosafety of NP is crucial for clinical relevance. In the future, we hope that a profound understanding of biological processes combined with novel material science technologies will contribute to successful clinical translations.

## Figures and Tables

**Figure 1 ijms-20-03490-f001:**
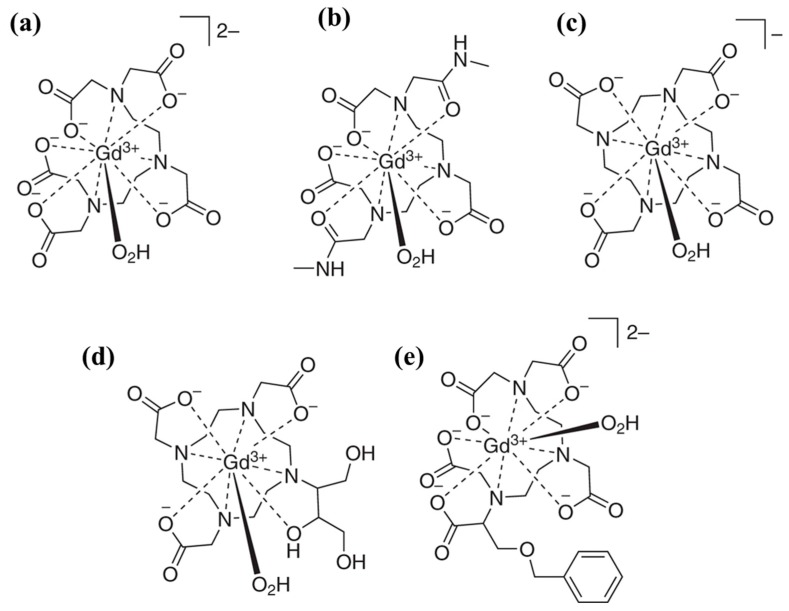
Chemical structures of five most commonly administered gadolinium-based contrast agents for clinical MRI: (**a**) Magnevist^®^, [Gd(OH_2_)(dtpa)]^2−^; (**b**) Omniscan^®^, [Gd(OH_2_)(dtpa-bma)]; (**c**) Dotarem^®^, [Gd(OH_2_)(dota)]^−^; (**d**) Gadovist^®^, [Gd(OH_2_)(do3a-butrol)]; and (**e**) Multihance^®^, [Gd(OH_2_)(bopta)]^2−^. Reproduced with permission from Reference [46]. Copyright 2019 Springer Nature.

**Figure 2 ijms-20-03490-f002:**
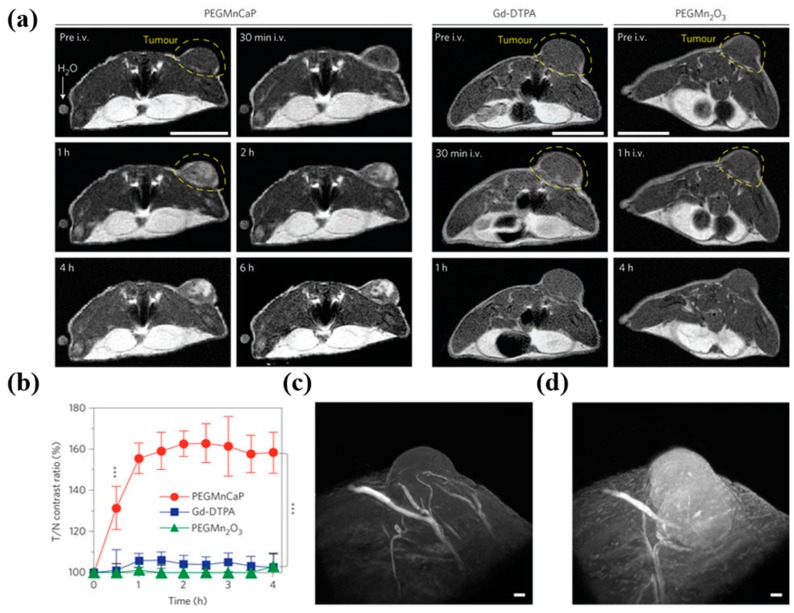
(**a**) In vivo MR images of subcutaneous C26 tumor-bearing mice pre- and post i.v. injection of PEGMnCaP (left), Gd-DTPA (center), and PEGMn_2_O_3_ (right) measured with 1 T MRI. Scale bar, 1 cm. (**b**) A comparison of the T/N contrast ratio after administration of PEGMnCaP (*n*  =  7), Gd-DTPA (*n*  =  5), and PEGMn_2_O_3_ (*n*  =  3). Three-dimensional MRI of C26 tumors before (**c**) and 1 h after (**d**) the i.v. injection of PEGMnCaP measured with 7 T MRI. Scale bars, 50 µm. Reproduced with permission from Reference [48]. Copyright 2016 Springer Nature.

**Figure 3 ijms-20-03490-f003:**
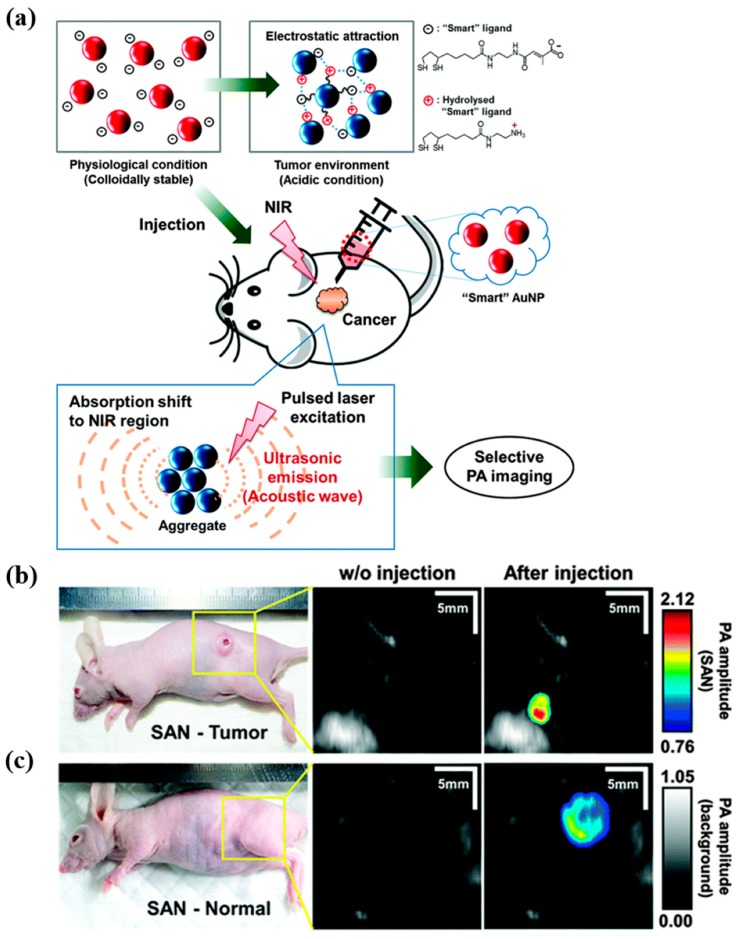
(**a**) Schematic illustration of the working mechanism of “smart” gold nanoparticles (SAN) for photo acoustic imaging (PAI). (**b**) Photograph (left) and PA MAP images (680-nm laser illumination) of tumor xenograft models before (center) and after (right) injecting SAN solution and (**c**) photograph (left) and PA MAP images of normal model before (center) and after (right) injecting SAN solution. Reprinted with permission from Reference [67]. Copyright 2016 Royal Society of Chemistry.

**Figure 4 ijms-20-03490-f004:**
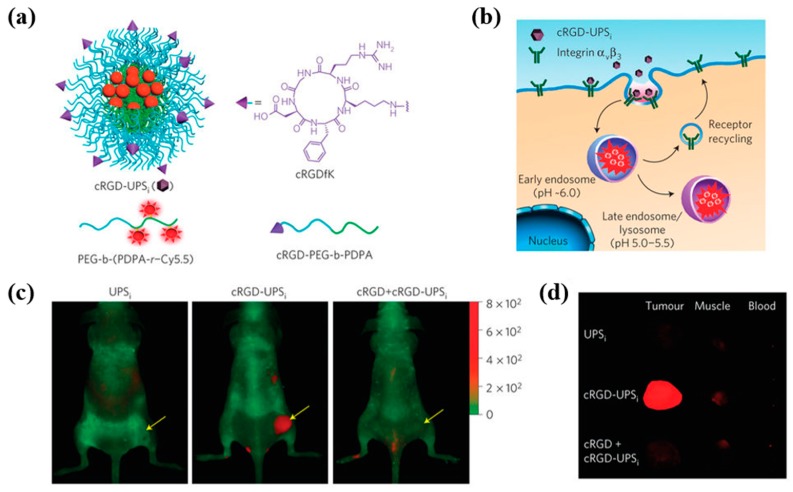
(**a**) Design of the cRGD-UPS_i_ nanoprobe. (**b**) Schematic illustration of α_v_β_3_-mediated endocytosis of the cRGD-UPS_i_ nanoprobe in tumor endothelial cells. (**c**) Superimposed fluorescent images of A549-tumor-bearing mice at 6 h postinjection of cRGD-UPS_i_ or the UPS_i_ nanoprobe. In the competition group, a blocking dose of cRGD peptide was injected 30 min before cRGD-UPS_i_ administration. Cy 5.5 (red) and autofluorescence (green) are separately shown in the composite images and (**d**) representative images of ex vivo tumors, muscles, and blood at 6 h post-injection of nanoprobes. Reprinted with permission from Reference [81]. Copyright 2014 Springer Nature.

**Figure 5 ijms-20-03490-f005:**
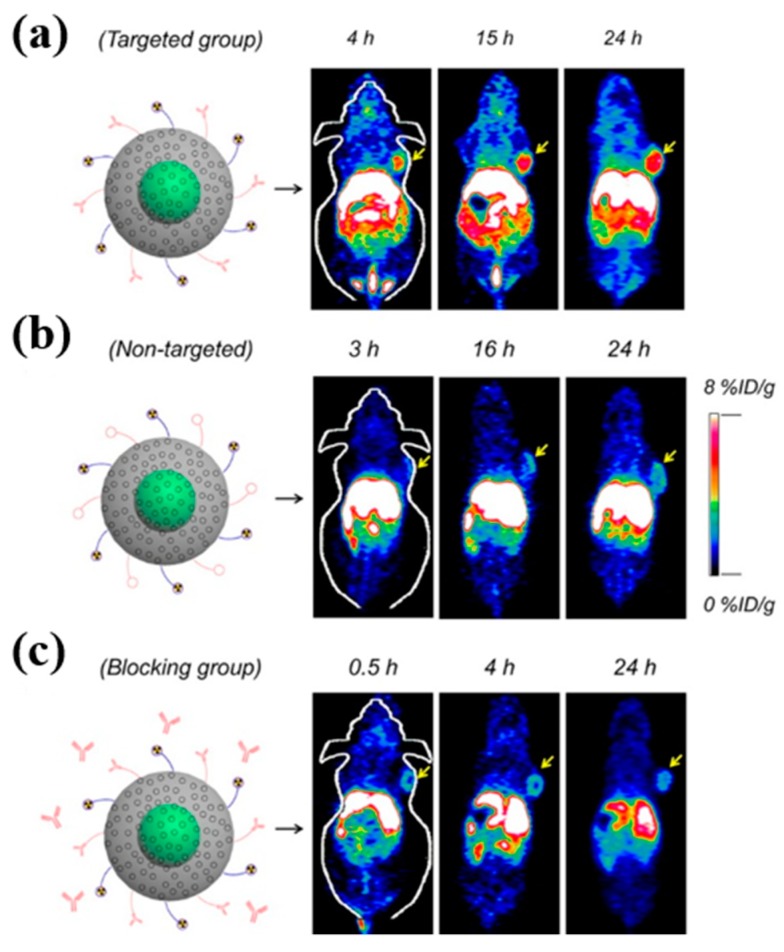
In vivo serial coronal PET images of (**a**) ^64^Cu-CuS@MSN-TRC105 nanoconjugates (targeted group), (**b**) ^64^Cu-CuS@MSN (non-targeted group), and (**c**) ^64^Cu-CuS@MSN-TRC105 with a large dose of free TRC105 (blocking group) in 4T1 murine breast tumor-bearing mice at different time points postinjection. Reprinted with permission from Reference [101]. Copyright 2015 American Chemical Society.

**Figure 6 ijms-20-03490-f006:**
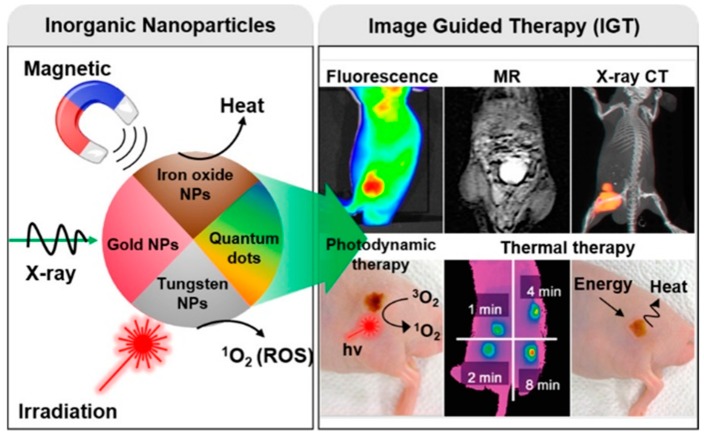
Inorganic NP for tumor imaging and therapy. Reproduced with permission from Reference [180]. Copyright 2017 American Chemical Society.

**Figure 7 ijms-20-03490-f007:**
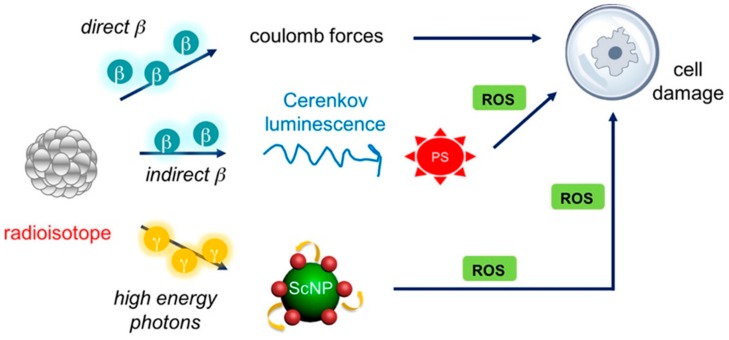
Radioisotopes can emit Cerenkov radiation to directly activate photosensitizer (PS) that can absorb light in the 200−500 nm range. Some radioisotopes can generate high-energy photons, such as γ-radiation, which can be absorbed by certain types of NP (i.e., scintillating NP). In this case, the PS will be indirectly excited by the radionuclide. Once the PS is activated, reactive oxygen species (ROS) will be generated to damage the surrounding cell/tissue. Reproduced with permission from Reference [216]. Copyright 2016 American Chemical Society.

**Table 1 ijms-20-03490-t001:** Salient features of clinical imaging modalities.

Imaging Modality	Principle	Temporal Resolution	Spatial Resolution	Penetration Depth	Sensitivity (M)
**PET**	Positrons	Seconds-minutes	1–2 nm	no limits	10^−11–^10^−12^
**SPECT**	low energy γ ray	Minutes	1–2 nm	no limits	10^−10–^10^−11^
**CT**	X-ray	Minutes	50–200 µm	no limits	10^−6^
**MRI**	Radio waves	Minutes-hours	25–100 µm	no limits	10^−9–^10^−6^
**Optical**	Near infrared light	Seconds-minutes	2–3 mm	<1 cm	10^−9–^10^−12^
**Photoacoustic**	Sound	Seconds-minutes	10 µm to 1 mm	6 mm-5 cm	Not determined

**Table 2 ijms-20-03490-t002:** NP-based contrast agents for imaging modalities.

Imaging Modality	NP	Size (nm)	Experiment Subject	Ref
**MRI (*T1*)**	MnO	7,15,20,25	MDA-MB 435	[47]
	PEG-MnCaP	60	C26	[48]
	MnO@Mn_3_O_4_	150–200	NIH3T6.7	[49]
	MnO	~50	HeLa cell	[50]
***T2***	Fe_3_O_4_	6.16	U87CD4.CXCR4	[55]
	Fe_3_O_4_	33.7	C6	[56]
	Fe_3_O_4_	~25	SK-BR-3, KB	[57]
	Fe_3_O_4_@MSN	75.3	U87MG	[58]
**PAI**	Semiconductor polymer	~23.84	4T1	[66]
	Au	~10.7	HeLa	[67]
	Self-assembled	135	U87MG	[73]
	IR775-Phe-Phe-Tyr(H_2_PO_3_)-OH	39.4	HeLa	[74]
**OPI**	Ag_2_S-QD	~6.12	4T1	[78]
	PEG-b-(PDPA-γ-Cy5.5)	<30	A549	[81]
	PEG-b-(PEPA-ICG)	~21.9	4T1	[82]
	NIR-II dye-PS-g-ICG	~12	4T1	[86]
**CT**	(BiO)_2_CO_3_	~1.3	Huh 7	[93]
	Iodine-dendrimer	~9	4T1	[94]
	Au	~3.7	BxPC3	[96]
**SPECT**	^99m^Tc-Fe_3_O_4_	~7.5	LS180 tumor	[98]
	^111^In-Au	10	A431 tumor	[99]
	^111^In-MSN	70	U87MG	[100]
**PET**	^64^Cu-CuS@MSN	~18.3	4T1 tumor	[101]
	^64^Cu-Au	~27	EMT-6 tumor	[104]
	^68^Ga-dendrimer	13.8	HT29, U87, 22Rv1, SOJ-6 tumor	[105]

**Table 3 ijms-20-03490-t003:** X-ray scintillators for cancer therapy.

Nanosytems	Size	PS	Loading Method	X-ray Dose	Exp. Subject	Ref
Tb_2_O_3_	10 nm	Porphyrin	Covalent	44 kV, 40 mA, 5.4 mGy/S	N/A	[219]
GdEuCl_2_ micelle	4.6 nm	Hyp	Physical	400 mA	Hela (*in vitro*)	[220]
LaF_3_:Tb	25–44 nm	RB	Pore	75 kV, 20 mA	N/A	[221]
Mo_6_I_8_	50 nm	Self	Physical	100 keV	N/A	[222]
MC540-Eu	400 nm	MC5 40	Pore	0.5 Gy, 50 kV	U87MG xenograft i.t	[223]
LiYF_4_:Ce@SiO_2_	50 nm	ZnO	Coating	8 Gy, 220 keV	HeLa xenograft i.t.	[224]
LaF_3_:Ce	2 µm	PPIX	Physical	2 Gy, 90 kV, 5 mA	PC-3 (in vitro)	[225]
MC540-Eu	400 nm	MC540	Pore	0.5 Gy, 50 kV	H1299 (in vitro and in vivo)	[226]
CeF_3_	7–11 nm	VP	Physical	6 Gy, 8 keV, 30 keV, 6 MeV	Panc 1 (in vitro)	[228]

C540-Eu (MC540-SAO:Eu@mSiO_2_), Mo_6_I_8_(n-Bu_4_N)_2_[Mo_6_I_8_(OOCC_10_H_15_)_6_], RB (rose bengal), PPIX (protoporphyrin IX), VP (verteporfin), HyP (hypericin), N/A: Not available.

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
