# Peer review of "Seeing Better and Going Deeper in Cancer Nanotheranostics"

_ijms, 2019, doi:10.3390/ijms20143490_

Round 1
Reviewer 1 Report
In the current manuscript, authors provide a review on nanomaterials for various imaging techniques. Although the manuscript is detailed, however it lacks overall direction and purpose. Also, the need for such a review wherein several current reviews are already available is not presented clearly. In addition, the following issues need to be addressed prior to accepting the manuscript for publication:
In Table 1, second(s) and minute(s) should be limited to one line, here (s) is coming in the second line.
Paragraph 2 comes right after Paragraph 1 without any introduction to imaging agents or therapeutic agents and starts talking about why current agents haven't been successful. A small paragraph should be added.
An introduction about various chemical agents current used for each of the imaging techniques discussed must be provided. Also, chemical structures of agents such as Gd-based MRI contrast agents should be provided for reference.
References are not updated. For example when talking about small molecule MRI contrast agents (ref 32, 33) are old and recent comprehensive reviews, eg. Wahsner et. al. Chem rev 2019 should be added. This is a recurrent theme and the manuscript needs updated references.
Authors mention that Manganese ions have emerged as safer alternative to Gd.. which is not entirely correct. Manganese based compounds is a better term, especially due to the toxicity of manganese oxide based materials.
Authors mention that Gd-based agents are known to induce NFD, however only open chain acyclic Gd-chelates are known to do that, whereas macrocyclic agents are very stable. This point should be specified.
For T2 based MRI contrast agents, no introduction is given and authors suddenly go to a specific example. An introduction to various T2 contrast agents should be given.
Similarly, during each separate section for each imaging technique, there is no or very minimal introduction about the type of agents that have been used and the reasons behind the choice. As this is a general review, considerable space must be given to why certain chemical agents are used beyond others and the reasons behind that choice.
As various agents and active elements under each imaging technique are mentioned, a table should be included in each section discussing the various elements used, as provided in the Image guided therapy section.
Author Response
The authors truly appreciate the reviewer’s valuable suggestions and comments. All the comments raised were carefully studied and reflected in the revised manuscript. Detailed responses to the comments are as follows:
Responses to Reviewer1:
1. In table 1, second(s) and minutes (s) should be limited to one line, here (s) is coming in the second line.
Response: Thanks for the comments. The error has been corrected in the revised manuscript.
2. Paragraph 2 comes right after paragraph 1 without any introduction to imaging agents or therapeutic agents and starts talking about why current agents haven’t been successful. A small paragraph should be added.
Response: We appreciate the reviewer for the important suggestion. The following paragraph was inserted in introduction part of the revised manuscript (Page 2).
“Vast advances in chemical and biological sciences resulted in the development of various therapeutic and imaging contrast agents. For instance, PET imaging can locate highly proliferative tumors using fluorodeoxyglucose ([18F]FDG) [22]. When administered they can preferentially enter high metabolic rate cells and are phosphorylated thereby accumulate at cancer site. Similarly, PAI can image B7-H3 overexpressing breast cancer using affibody conjugated clinically approved near infrared (NIR) dye indocyanine green [23]. Optical imaging fluorescent dyes that can emit strong fluorescence signal in the presence of pathogenic stimuli were developed such as pH probes, oxygen sensitive probes, bioreductive and activity based probes [24].”
3. An introduction about various chemical agents currently used for each of the imaging techniques discussed must be provided. Also, chemical structures of agents such as Gd-based MRI contrast agents should be provided for reference.
Response: Thanks for the suggestion. Introduction to imaging contrast agents used in various modalities has been added and described in the revised manuscript and chemical structure of Gd based contrast agents
(Figure 1) were provided in the revised manuscript.
T1 contrast agents (page 3)
Gadolinium ions (Gd3+) are the traditionally used excellent T1 contrast agent due to their ability to generate bright signals by shortening the T1 time (Figure 1), and hence several formulations have been approved by the food and drug administration U.S. (FDA) for clinical use [38, 39]. Open chain acyclic Gd-chelates when administered, has a tendency to leach and expose toxic heavy metal, which is known to induce nephrogenic system fibrosis (NSF) in patients [41, 42], so very stable macrocyclicGd-chelates are currently under investigation. In this juncture, manganese compounds emerge as a safe alternative T1 contrast agent. It does not induce renal toxicity like Gd3+, hence numerous Mn-based nanoparticles have been developed as preclinical in vivo T1 agents [43, 45].
T2 contrast agents (page 6)
The most common T2 contrast agent is super paramagnetic iron oxide NP (SPION), high magnetization exhibited by iron oxide causes magnetic inhomogeneities affecting T2 relaxation times. Briefly, dipolar interactions between iron oxide magnetic moment and water proton spins decreases the T2 relaxation times leading to negative image contrast[53]. There are several commercially available T2 contrast agents in the market namely, ferumoxtron (Sinerem (EU), Combidex (US)), ferumoxytol (Faraheme (US)) and ferumoxide (Senti-Scint Feridex (US), Endorem (Britain)).
PAI contrast agents (page 7)
Among Au nanomaterials, gold nanorods (GNR) has gained significant attention as PAI agent due to its tunable NIR absorption and facile synthesis. Cylinder like morphology of GNR affects absorption band and absorption shift towards NIR region with increase in the particles length to width ratio. Hence, better PAI performance can be achieved by adjusting the aspect ratio. Other nanomaterial based on gold used for PAI is gold nanocages (AuNCs), these are cubic NP with a hollow nanostructure with an absorption band ranging from 600-1200 nm can be an excellent PAI agent for biological applications. Semiconducting NP that consists of semiconducting polymer and oligomer exhibits high photostabilty, large absorption coefficient, tunable optical absorption and controllable size, which can serve as a PAI agent [65].
OPI contrast agents (page 10)
Quantum dots (QDs) are inorganic NP with excellent optical properties have been widely used in biomedical imaging, diagnostic and therapeutic applications. Due to the quantum confinement, QDs exhibit size dependent optical and electronic properties. QDs presents advantages over conventional organic dyes such as, lower photobleaching and size dependent photoluminescence emission. Au NP plays an integral role in biomedical field due to their ease of synthesis and tunable optical and electronic properties [76]. The unique properties of Au NP arise from their surface plasmon resonance. The surface of AuNP can be easily modified with proteins, peptides, oligonucleotides, and many other compounds, while still maintaining their optical properties. The advantages of AuNP over their metallic counterpart are their chemical inertness and reduced toxicity [77].
Nuclear imaging contrast agents (page 12)
[18F] FDG is the primary agent used for the detection and staging of many cancers using PET. [18F] FDG targets tumor with high metabolism rate and used in 90% of human PET studies [19]. 64Cu is a promising radionuclide with suitable decay (half-life = 12.7 h) characteristics and known for the convenience with which it can be used for radiolabelling. In addition, 64Cu can be readily prepared with high activity by biomedical cyclotron. For SPECT imaging, 99mTc radionuclide is a metastable isomer of 99Tc, widely used for diagnostic imaging with a half-life of 6.0 h. 111In is also a SPECT imaging agent useful for isotopic labelling of biomolecules for specialized diagnostic applications [88,89].
Due to their large X-ray absorption coefficient, Au NP has been demonstrated as an effective CT contrast agent compared to traditional iodine-based contrast agents. Gold exhibits much higher X-ray absorption coefficient than iodine at 100 keV [90]. Surface modifications of Au NP can enhance absorption coefficient, for instance, attenuation coefficient achieved by PEG-coated GNPs to be 5.7 times higher than current iodine-based CT contrast agents [91]. Bismuth-based NP as CT contrast agent presents advantages over conventional iodinated and Au-based NP contrast agents due to their high atomic number, and X-ray attenuation coefficent, low cost, and low toxicity. Bismuth sulfide holds great promise as a NP contrast agent due to its high effective nuclear charge, physical density, and electron density, which makes them an excellent candidate for CT imaging [92].
4. References are not updated. For example when talking about small molecules MRI contrast agents (ref 32,33) are old and recent comprehensive reviews, eg. Wahsner et al. Chem rev 2019 should be added. This is a recurrent theme and the manuscript needs updated references.
Response: Thanks for the reviewer’s comments. The recommended review and following references have been introduced in the revised manuscript.
1. Seeram, E. Computed tomography: A technical review. Radiol. Technol. 2018, 89, 279-302.
2. Ginat, G.T.; Gupta, R. Advances in computed tomography imaging technology. Annu. Rev. Biomed. Eng. 2014, 16, 431-453.
3. Van Straten, D.; Mashayekhi, V.; De Brujin, H.S.; Oliveria, S.; Robinson, D.J. Oncologic photodynamic therapy: Basic principles, current clinical status and future directions.Cancers (Basel). 2017, 9, 1-54.
4. Kelloff, G.J.; Hoffman, J.M.; Johnson, B.; Scher, H.I.; Siegel, B.A.; Cheng, E.Y.; Cheson, B.D.; O'Shaughnessy, J.; Guyton, K.Z.; Mankoff, D.A.; Shankar, L.; Larson, S.M.; Sigman, C.C.; Schilsky, R.L.; Sullivan, D.C. Progress and Promise of FDG-PET Imaging for Cancer Patient Management and Oncologic Drug Development Clin. Cancer. Res. 2006, 11, 2785-2808.
5. Bam, R.; Laffey, M.; Nottberg, K.; Lown, P.S.; Hackel, B.J.; Wilson, K.E. Affibody-indocyanine green based contrast agent for photoacoustic and fluorescence molecular imaging of B7–H3 expression in breast cancer Bioconjugate Chem. 2019, 30, 1677-1689.
6. Wu, Y.; Zhang, W.; Li, J.; Zhang, Y. Optical imaging of tumor microenvironment Am. J. Nucl. Med. Mol. Imaging. 2013, 3, 1-15.
Washner, J.; Gale, E.M.; Rodriguez-Rodriguez, A.; Caravan, P. Chemistry of MRI contrast agents: Current challenges and new frontiers. Chem.Rev.2019, 119, 957-1057.
7. Clough, T.J.; Jiang, L.; Wong, K.L.; Long, N.J. Ligand design strategies to increase stability of gadolinium-based magnetic resonance imaging contrast agents. Nat. Commun2019, 10, 1-14.
8. Wang. J.; Wang, H.; Ramsay, I.A.; Erstad, D.J.; Fuchs, B.C.; Tanabe, K.K.; , Caravan, P.; Gale, E.M. Manganese-Based Contrast Agents for Magnetic Resonance Imaging of Liver Tumors: Structure-Activity Relationships and Lead Candidate Evaluation. J.Med.Chem2018, 61, 8811-8824
9. Lee, N.; Yoo, D.; Ling, D.; Cho, M.H.; Hyeon, T. Iron oxide based nanoparticles for multimodal imaging and magnetoresponsive therapy Chem. Rev. 2015, 115, 10637-10689
10. Fu, Q.; Zhu, R.; Song, J.; Yang, H.; Chen, X. Photoacoustic Imaging: Contrast Agents and Their Biomedical Applications. Adv. Mater. 2019, 31, 1-31.
11. Ji, X.; Peng, F.; Zhong, Y.; Su, He, Y. Fluorescent quantum dots: Synthesis, biomedical optical imaging, and biosafety assessment. Colloids Surf. B Biointerfaces. 2014, 124, 132-139.
12. Wu, Y.; Ali, M.R.K.; Chen, K.; Fang, N.; El-Sayed, M. Gold nanoparticles in biological optical imaging. Nanotoday2019, 24, 120-140.
13. Frazin, L.; Sheibani, S.; Moassesi, M.E.; Shamsipur, M. An overview of nanoscale and radionuclides and radiolabeled nanomaterials commonly used for nuclear molecular imaging and therapeutic functions. J. Biomed. Mater. Res. A. 2019, 107, 251-285.
14. Xing, Y.; Zhao, J.; Conti, P.S.; Chen, K. Radiolabeled nanoparticles for multimodality tumor imaging. Theranosctics. 2014, 4, 290-306.
15. Xi, D.; Dong, S.; Meng, X.; Lu, Q.; Meng, L.; Ye, J. Gold nanoparticles as computerized tomography (CT) contrast agents. RSC Adv. 2012, 2, 12515-12524.
16. Kim, D.; Park, S.; Lee, J.H.; Jeong, Y.Y.; Jon, S. Antibiofouling polymer-coated gold nanoparticles as a contrast agent for in vivo X-ray computed tomography imaging. J. Am. Chem. Soc. 2007, 129, 7661-7665.
17. Yeh, B.M.; FitxGeral, P.F.; Edic, P.M.; Lambert, J.W.; Colborn, R.E.; Marino, M.E.; Evans, P.M.; Roberts, J.C.; Wang, Z.J.; Wong, M.J.; Bonitatibus Jr, P.J. Opportunities for new CT contrast agents to maximize the diagnostic potential of emerging spectral CT technologies. Adv. Drug Deliv. Rev2017, 113, 201-222.
18. Moore, C.; Jokerst, J.V Strategies for Image-Guided Therapy, Surgery, and Drug Delivery Using Photoacoustic Imaging Theranostics2019, 9, 1550-1571.
Doughty, A.C.V.; Hoover, A.R.; Layton, E.; Murray, C.K.; Howardand, E.W.; Chen, W.R Nanomaterial Applications in Photothermal Therapy for Cancer Materials2019, 12, 1-14.
19. Tynga, I.M.; Abrahamse, H Nano-Mediated Photodynamic Therapy forCancer: Enhancement of Cancer Specificity andTherapeutic EffectsNanomaterials 2018, 8, 1-14
20. Duan, C.; Liang, L.; Li, L.; Zhang, R.; Xu, Z.P Recent progress in upconversion luminescence nanomaterials for biomedical applicationsJ. Mater. Chem. B2018, 6, 192-209.
21. Larue, L.; Ben Mihoub, A.; Youssef, Z.; Colombeau, L.; Acherar, S.; André, J.C.; Arnoux, P.; Baros, F.; Vermandel, M.; Frochot, CUsing X-rays in photodynamic therapy: an overviewPhotochem. Photobiol. Sci2018, 17, 1612-1650.
22. Chen, X.; Song, J.; Chen, X .; Huanghao, Yang X-ray-activated nanosystems for theranostic applications Chem. Soc. Rev 2019, 48, 3073-3101.
5. Authors mention that Manganese ions have emerged as safer alternative to Gd.. which is not entirely correct. Manganese based compounds is a better term, especially due to the toxicity of manganese oxide based materials.
Response: We appreciate the reviewer’s suggestion. Accordingly, the following sentence was inserted in the revised manuscript (page 3, line 102).
“Manganese compounds emerge as a safe alternative T1 contrast agent.”
6. Authors mention that Gd-based agents are known to induce NFD, however only open chain acyclic Gd-chelates are known to do that, whereas macrocyclic agents are very stable. This point should be specified.
Response: Thanks indeed for the suggestion. The point suggested by the reviewer has been specified in the revised manuscript.
Specified sentence (page 3, line 99)
“ Open chain acyclic Gd-chelates when administered, has a tendency to leach and expose toxic heavy metal, which is known to induce nephrogenic system fibrosis (NSF) in patients [31, 32], so very stable macrocyclic Gd-chelates are currently under investigation.”
7. For T2 based MRI contrast agents, no introduction is given and authors suddenly go to a specific example. An introduction to various T2 contrast agents should be given.
Response: Following the reviewer’s suggestion, the following paragraph was inserted in the revised manuscript.
T2 contrast agents (page 6)
The most common T2 contrast agent is super paramagnetic iron oxide NP (SPION), high magnetization exhibited by iron oxide causes magnetic inhomogeneities affecting T2 relaxation times. Briefly, dipolar interactions between iron oxide magnetic moment and water proton spins decreases the T2 relaxation times leading to negative image contrast [53]. There are several commercially available T2 contrast agents in the market namely, ferumoxtron (Sinerem (EU), Combidex (US)), ferumoxytol (Faraheme (US)) and ferumoxide (Senti-Scint Feridex (US), Endorem (Britain)).
8. Similarly, during each separate sections for each imaging technique, there is no or very minimal introduction about the type of agents that have been used and the reasons behind the choice. As this is a general review, considerable space must be given to why certain chemical agents are used beyond others and the reasons behind that choice.
Response: We appreciate the reviewer’s comments. We have combined the suggestions and revised accordingly in the revised manuscript. Please see the Response to Comments 3 for detailed description.
9. As various agents and active elements under each imaging technique are mentioned, a table should be included in each section discussing the various elements used, as provided in the image guided therapy section.
Response: Following the reviewer’s suggestion, Table 2 was inserted in the revised manuscript.
________________________________________________________________

Reviewer 2 Report
Referee Report
Manuscript number: ijms-538680
Title: Seeing Better and Going Deeper of Cancer Nanotheranostics
By Sivasubramanian et al
Submitted to IJMS
Comments:
This review is about using nanoparticles as imaging contrast agent in diagnosis and therapy. This work is quite comprehensive and thorough and is up to the publication level of IJMS. I only have some comments to further improve the presentation and quality of the manuscript:
1. Table 1: please rotate the table 90 degree to make it horizontal on the page.
2. Section 2: Please create a subsection to review the computed tomography (CT).
3. L564-573: Please quote the paper: Albayedh et al (J Med Phys 2018;43:195) in this paragraph as it is an important work on nanoparticle imaging contrast on kV imaging.
4. It is good to prepare a Table to show the recent progresses of contrast agent development for different imaging modalities with corresponding references.
Author Response
Responses to Reviewer 2
This review is about using nanoparticles as imaging contrast agent in diagnosis and therapy. This work is quite comprehensive and through and is up to the publication level of IJMS. I only have some comments to further improve the presentation and quality of the manuscript.
1. Table 1. Please rotate the table 90 degree to make it horizontal on the page.
Response: We truly appreciate the reviewer’s support. Following the suggestion, Table 1 has been modified and presented horizontally in the revised manuscript.
2. Section 2: Please create a subsection to review the computed tomography (CT).
Response: Thanks indeed for the suggestion. Accordingly, a subsection about CT was introduced in the review manuscript as follows:
CT imaging (page 12 and 13)
Renal clearable ultrasmall bismuth subcarbonate nanoclusters were prepared for tumor specific CT imaging. These nanoclusters can be assembled to bismuth subcarbonate nanotubes (BNTs), this unique structure allowed tumor specific accumulation followed by disassembly under acidic pH. In addition, DOX loaded BNTs exhibited excellent therapeutic efficacy when combined with radiotherapy [93]. Activity based probes specific to cathepsins were attached to iodinated polymeric dendrimers for the detection of solid tumors using CT imaging. Tumor specific accumulation of probes enabled CT imaging by activity dependent covalent binding. In addition, signal detection was achieved using a low dose of 20 mgI/kg compared to clinical dose of iodinated agents (300 mgI/kg) [94]. Albayedh et al. studied imaging contrast enhancement in radiotherapy by finding relationship between imaging contrast ratio and different parameters such as various NP (gold, iodine, silver, iron oxide and platinum) concentration, different beam energies for the different NP concentrations, various beam energies for Au NP, different thicknesses of the incident layer of the phantom including variety of gold NP concentration. Monte Carlo simulation results showed that Au NP had the highest imaging contrast ratio. In addition, it was proved that higher contrast will be obtained with high concentration of NP, low beam energy and small thickness of the tumor [95].
3. L564-573: Please quote the paper. Albayedh et al (J Med Phys 2018;43;195) in this paragraph as it is an important work on nanoparticle imaing contrast on kV imaging.
Response: Thanks for the reviewer’s comments. The suggested research article has been included under CT imaging section in the revised manuscript (page 13, line 411).
Albayedh et al. studied imaging contrast enhancement in radiotherapy by finding relationship between imaging contrast ratio and different parameters such as various NP (gold, iodine, silver, iron oxide and platinum) concentration, different beam energies for the different NP concentrations, various beam energies for Au NP, different thicknesses of the incident layer of the phantom including variety of gold NP concentration. Monte Carlo simulation results showed that Au NP had the highest imaging contrast ratio. In addition, it was proved that higher contrast will be obtained with high concentration of NP, low beam energy and small thickness of the tumor [95].
4. It is good to prepare a Table to show the recent progresses of contrast agent development for different imaging modalities with corresponding references.
Response: We sincerely appreciate the reviewer’s constructive suggestion. Accordingly, Table 2 has been incorporated in the revised manuscript.
________________________________________________________________
Again, we are very grateful to the reviewers for their helpful comments and constructive suggestions to make this review article more informative and comprehensive. Thanks a lot indeed.

Round 2
Reviewer 1 Report
The authors have addressed major concerns as pointed out in the earlier review. There are still a few typo and grammatical punctuation errors that should be corrected before accepting.
Author Response
Reviewer 1
Comments
The authors have addressed major concerns as pointed out in the earlier review. There are still a few typo and grammatical punctuation errors that should be corrected before accepting.
Response: Manuscript was carefully read, typos and grammatical errors are corrected in the revised manuscript. Please check the revised manuscript with track changes in the attachment. Thank you so much.

Reviewer 2 Report
The presentation and quality of this revision are improved. I am satisfied with the responses and corrections from the Authors.
Author Response
Reviewer 2
Comments
The presentation and quality of this revision are improved. I am satisfied with the responses and corrections from the Authors.
Response: We are very grateful to the reviewers for their helpful comments and constructive suggestions to make this review article more informative and comprehensive. Thanks a lot indeed.
